# UNDERSTANDING OVER-SQUASHING AND BOTTLENECKS ON GRAPHS VIA CURVATURE

**Jake Topping**[12†], **Francesco Di Giovanni**[3†], **Benjamin P. Chamberlain**[3],
**Xiaowen Dong**[1], **and Michael M. Bronstein**[23]
[1]University of Oxford  [2]Imperial College London  [3]Twitter

## ABSTRACT

Most graph neural networks (GNNs) use the message passing paradigm, in which node features are propagated on the input graph. Recent works pointed to the distortion of information flowing from distant nodes as a factor limiting the efficiency of message passing for tasks relying on long-distance interactions. This phenomenon, referred to as 'over-squashing', has been heuristically attributed to graph bottlenecks where the number of $k$-hop neighbors grows rapidly with $k$. We provide a precise description of the over-squashing phenomenon in GNNs and analyze how it arises from bottlenecks in the graph. For this purpose, we introduce a new edge-based combinatorial curvature and prove that negatively curved edges are responsible for the over-squashing issue. We also propose and experimentally test a curvature-based graph rewiring method to alleviate the over-squashing.

## 1 INTRODUCTION

In the past few years, deep learning on graphs and in particular graph neural networks (GNNs) (Sperduti, 1994; Goller & Kuchler, 1996; Sperduti & Starita, 1997; Frasconi et al., 1998; Gori et al., 2005; Scarselli et al., 2008; Bruna et al., 2014; Defferrard et al., 2016; Kipf & Welling, 2017; Gilmer et al., 2017) have become very popular in the machine learning community due to their ability to deal with broad classes of systems of relations and interactions, ranging from social networks to particle physics (Shlomi et al., 2021).

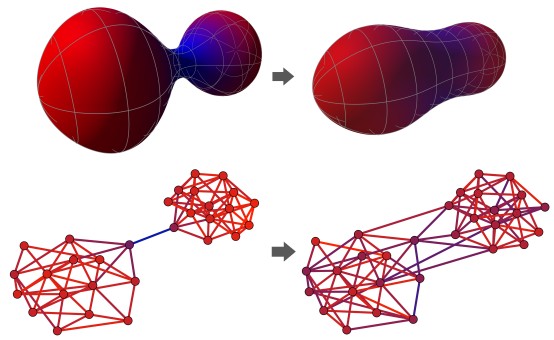

Figure 1: Top: evolution of curvature on a surface may reduce the bottleneck. Bottom: this paper shows how the same may be done on graphs to improve GNN performance. Blue/red shows negative/positive curvature.

The vast majority of GNNs follow the *message passing* paradigm (Gilmer et al., 2017), using learnable non-linear functions to diffuse information on the graph. Multiple popular GNN architectures such as GCN (Kipf & Welling, 2017) and GAT (Veličković et al., 2018) can be posed as particular flavors of this scheme and considered instances of a more general framework of geometric deep learning (Bronstein et al., 2021).

Some of the drawbacks of the message passing paradigm have now been identified and formalized, including the limits of expressive power (Xu et al., 2019; Morris et al., 2019; Maron et al., 2019) and the problem of over-smoothing (NT & Maehara, 2019; Oono & Suzuki, 2020). On the other hand, much less is known about the phenomenon of *over-squashing*, consisting in the distortion of messages being propagated from distant nodes. Alon & Yahav (2021) proposed rewiring the graph as a way of reducing the *bottleneck*, defined as those topological properties in the graph leading to over-squashing. This approach is in line with multiple other results e.g. using connectivity diffusion (Klicpera et al., 2019) as a preprocessing step to facilitate graph learning. Yet, the exact understanding of the over-squashing and how it originates from the bottlenecks in the topology of the underlying

---

[†]Equal contribution

graph are still elusive. Consequently, there is currently no consensus on the right method (either based on graph rewiring or not) to address the bottleneck and hence alleviate the over-squashing.

In this paper, we address these questions using tools from differential geometry, which traditionally is concerned with the study of manifolds. It offers an appealing framework to study the properties of graphs, in particular arguing that graphs, like manifolds, exhibit *curvature* that makes them more suitable to be realized in spaces with hyperbolic geometry (Liu et al., 2019; Chami et al., 2019; Boguna et al., 2021). One notion of curvature that has received attention for graph learning is *Ricci curvature* (Hamilton, 1988), also known in geometry for its use in Ricci flow and the subsequent proof of the Poincaré conjecture (Perelman, 2003). Certain graph analogues of the Ricci curvature (Forman, 2003; Ollivier, 2009; Sreejith et al., 2016) were used in Ni et al. (2018) for a discrete version of Ricci flow to construct a metric between graphs. Graph Ricci flow was also used in Ni et al. (2019) for community detection. Both of these methods use the edge weights as a substitute for the metric of a manifold, and do not change the topological structure of the graph.

**Contributions and Outline.** This paper, to our knowledge, is the first theoretical study of the bottleneck and over-squashing phenomena in message passing neural networks from a geometric perspective. In Section 2, we propose the Jacobian of node representations as a formal way of measuring the over-squashing and we show that the graph topology may compromise message propagation in graph neural networks by creating a bottleneck. In Section 3, we investigate how such a bottleneck is induced which leads to the over-squashing of information. To this aim, we introduce a new combinatorial edge-based curvature called Balanced Forman curvature that constitutes a sharp lower bound to the standard Ollivier curvature on graphs, and prove that negatively curved edges are responsible for the formation of bottlenecks (and hence for over-squashing). In Section 4, we present a new curvature-based method for graph rewiring called Stochastic Discrete Ricci Flow. According to the theoretical results in Section 3, this rewiring method is suited to address the graph bottleneck and hence alleviate the over-squashing by surgically targeting the edges responsible for the issue. By contrast, we rigorously show that a recently introduced diffusion-based rewiring scheme might generally fail to reduce the bottleneck. Finally, in Section 5, we compare different rewiring strategies experimentally on several standard graph learning datasets.

## 2 ANALYSIS OF THE OVER-SQUASHING PHENOMENON

### 2.1 PRELIMINARIES

Let $G = (V, E)$ be a simple, undirected, and connected graph, where $(i, j) \in E$ iff $i \sim j$. We focus on the unweighted case, although the theory extends to the weighted setting as well. We denote the adjacency matrix by $A$ and let $\tilde{A} = A + I$ be the adjacency matrix augmented with self-loops. Similarly we let $\tilde{D} = D + I$, with $D$ the diagonal degree matrix, and let $\hat{A} = \tilde{D}^{-\frac{1}{2}} \tilde{A} \tilde{D}^{-\frac{1}{2}}$ be the normalized augmented adjacency matrix (self-loops are commonly included in GNN architecture, and in Section 2 we formally explain why GNNs are expected to propagate information more reliably when self-loops are taken into account). Given $i \in V$, we denote its degree by $d_i$ and let

$$S_r(i) := \{j \in V : d_G(i, j) = r\}, \quad B_r(i) := \{j \in V : d_G(i, j) \leq r\},$$

where $d_G$ is the standard shortest-path distance on the graph and $r \in \mathbb{N}$. The set $B_r(i)$ represents the *receptive field* of an $r$-layer message passing neural network at node $i$.

**Message passing neural networks (MPNNs).** Assume that the graph $G$ is equipped with node features $X \in \mathbb{R}^{n \times p_0}$ where $x_i \in \mathbb{R}^{p_0}$ is the feature vector at node $i = 1, \ldots, n = |V|$. We denote by $h_i^{(\ell)} \in \mathbb{R}^{p_\ell}$ the representation of node $i$ at layer $\ell \geq 0$, with $h_i^{(0)} = x_i$. Given a family of message functions $\psi_\ell : \mathbb{R}^{p_\ell} \times \mathbb{R}^{p_\ell} \to \mathbb{R}^{p'_\ell}$ and update functions $\phi_\ell : \mathbb{R}^{p_\ell} \times \mathbb{R}^{p'_\ell} \to \mathbb{R}^{p_{\ell+1}}$, we can write the $(\ell + 1)$-st layer output of a generic MPNN as follows (Gilmer et al., 2017):

$$h_i^{(\ell+1)} = \phi_\ell \left( h_i^{(\ell)}, \sum_{j=1}^{n} \hat{A}_{ij} \psi_\ell(h_i^{(\ell)}, h_j^{(\ell)}) \right). \tag{1}$$

Here we have used the augmented normalized adjacency matrix to propagate messages from each node to its neighbors, which simply leads to a degree normalization of the message functions $\psi_\ell$. To

avoid heavy notations the node features and representations are assumed to be scalar from now on; these assumptions simplify the discussion and the vector case leads to analogous results.

## 2.2 THE OVER-SQUASHING PROBLEM

Multiple recent papers observed that MPNNs tend to perform poorly in situations when the learned task requires long-range dependencies and at the same time the structure of the graph results in exponentially many long-range neighboring nodes. We say that a graph learning problem has *long-range dependencies* when the output of a MPNN depends on representations of distant nodes interacting with each other. If long-range dependencies are present, messages coming from non-adjacent nodes need to be propagated across the network without being too distorted. In many cases however (e.g. in 'small-world' graphs such as social networks), the size of the receptive field $B_r(i)$ grows exponentially with $r$. If this occurs, representations of exponentially many neighboring nodes need to be compressed into fixed-size vectors to propagate messages to node $i$, causing a phenomenon referred to as *over-squashing* of information (Alon & Yahav, 2021). In line with Alon & Yahav (2021), we refer to those structural properties of the graph that lead to over-squashing as a *bottleneck*[1].

**Sensitivity analysis.** The hidden feature $h_i^{(\ell)} = h_i^{(\ell)}(x_1, \ldots, x_n)$ computed by an MPNN with $\ell$ layers as in equation 1 is a differentiable function of the input node features $\{x_1, \ldots, x_n\}$ as long as the update and message functions $\phi_\ell$ and $\psi_\ell$ are differentiable. The over-squashing of information can then be understood in terms of one node representation $h_i^{(\ell)}$ failing to be affected by some input feature $x_s$ of node $s$ at distance $r$ from node $i$. Hence, we propose the *Jacobian* $\partial h_i^{(r)} / \partial x_s$ as an explicit and formal way of assessing the over-squashing effect[2].

**Lemma 1.** *Assume an MPNN as in equation 1. Let $i, s \in V$ with $s \in S_{r+1}(i)$. If $|\nabla \phi_\ell| \leq \alpha$ and $|\nabla \psi_\ell| \leq \beta$ for $0 \leq \ell \leq r$, then*

$$\left| \frac{\partial h_i^{(r+1)}}{\partial x_s} \right| \leq (\alpha\beta)^{r+1} (\hat{A}^{r+1})_{is}. \tag{2}$$

Lemma 1 states that if $\phi_\ell$ and $\psi_\ell$ have bounded derivatives, then the propagation of messages is controlled by a suitable power of $\hat{A}$. For example, if $d_G(i, s) = r + 1$ and the sub-graph induced on $B_{r+1}(i)$ is a binary tree, then $(\hat{A}^{r+1})_{is} = 2^{-1}3^{-r}$, which gives an exponential decay of the node dependence on input features at distance $r$, as also heuristically argued by Alon & Yahav (2021).

The sensitivity analysis in Lemma 1 relates the over-squashing – as measured by the Jacobian of the node representations – to the graph topology via powers of the augmented normalized adjacency matrix. In the next section we explore this connection further by analyzing which local properties of the graph structure affect the right hand side in equation 2, hence causing the bottleneck. We will address this problem by introducing a new combinatorial notion of edge-based curvature and showing that *negatively curved edges are those responsible for the over-squashing phenomenon*.

## 3 GRAPH CURVATURE AND BOTTLENECK

A natural object in Riemannian geometry is the *Ricci curvature*, a bilinear form determining the geodesic dispersion, i.e. whether geodesics starting at nearby points with 'same' velocity remain parallel (*Euclidean space*), converge (*spherical space*), or diverge (*hyperbolic space*). To motivate the introduction of a Ricci curvature for graphs, we focus on these three cases. Consider two nodes $i \sim j$ and two edges starting at $i$ and $j$ respectively. In a discrete spherical geometry (Figure 2a), the edges would meet at $k$ to form a triangle (complete graph). In a discrete Euclidean geometry (Figure 2b), the edges would stay parallel and form a 4-cycle based at $i \sim j$ (orthogonal grid). Finally, in a discrete hyperbolic geometry (Figure 2c), the mutual distance of the edge endpoints would have grown compared to that of $i$ and $j$ (tree). Therefore, a Ricci curvature for graphs should provide us with more sophisticated tools than the degree to analyze the neighborhood of an edge.

---

[1] We note that the over-squashing issue is different from the problem of under-reaching; the latter simply amounts to a MPNN failing to fully explore a graph when the depth is smaller than the diameter (Barceló et al., 2019). The over-squashing phenomenon instead may occur even in deep GNNs with the number of layers larger than the graph diameter, as tested experimentally in Alon & Yahav (2021).

[2] The Jacobian of a GNN-output was also used by Xu et al. (2018) to set a similarity score among nodes.

**Curvatures on graphs.** The main examples of edge-based curvature are the *Forman curvature* $F(i,j)$ (Forman, 2003) and the *Ollivier curvature* $\kappa(i,j)$ in Ollivier (2007; 2009) (see Appendix). While $F(i,j)$ is given in terms of combinatorial quantities (Sreejith et al., 2016), results are scarce and the definition is biased towards negative curvature. The theory on $\kappa(i,j)$ instead is richer (Lin et al., 2011; Münch, 2019) but its formulation makes it hard to control local quantities.

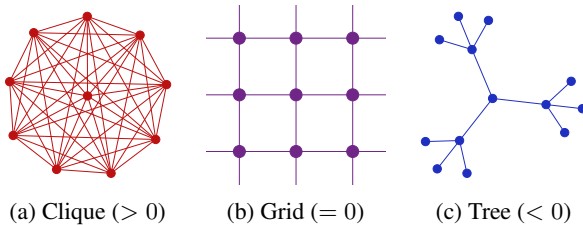

(a) Clique ($> 0$)     (b) Grid ($= 0$)     (c) Tree ($< 0$)

Figure 2: Different regimes of curvatures on graphs analogous to spherical (a), planar (b), and hyperbolic (c) geometries in the continuous setting.

**Balanced Forman curvature.** We propose a new curvature to address the shortcomings of the existing candidates. We use the following definitions to describe the neighborhood of an edge $i \sim j$ and we refer to the Appendix for a more complete discussion:

(i) $\sharp_\Delta(i,j) := S_1(i) \cap S_1(j)$ are the triangles based at $i \sim j$.

(ii) $\sharp_\square^i(i,j) := \{k \in S_1(i) \setminus S_1(j), k \neq j : \exists w \in (S_1(k) \cap S_1(j)) \setminus S_1(i)\}$ are the neighbors of $i$ forming a 4-cycle based at the edge $i \sim j$ *without* diagonals inside.

(iii) $\gamma_{\max}(i,j)$ is the maximal number of $4-$cycles based at $i \sim j$ traversing a common node (see Definition 4).

In line with the discussion about geodesic dispersion, one expects $\sharp_\Delta$ to be related to positive curvature (complete graph), $\sharp_\square^i$ to zero curvature (grid), and the remaining *outgoing* edges to negative curvature (tree). Our new curvature formulation reflects such an intuition and recovers the expected results in the classical cases. In the example in Figure 3 we have $\sharp_\square^0(0,1) = \{2,3\}$ while $\sharp_\square^1(0,1) = \{5\}$, both without 4,6 because of the triangle 1-6-0. The degeneracy factor $\gamma_{\max}(0,1) = 2$, as there exist two 4-cycles passing through node 5.

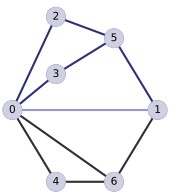

Figure 3: 4-cycle contribution.

**Definition 1 (Balanced Forman curvature).** *For any edge $i \sim j$ in a simple, unweighted graph $G$, we let* $\mathrm{Ric}(i,j)$ *be zero if* $\min\{d_i, d_j\} = 1$ *and otherwise*

$$\mathrm{Ric}(i,j) := \frac{2}{d_i} + \frac{2}{d_j} - 2 + 2\frac{|\sharp_\Delta(i,j)|}{\max\{d_i,d_j\}} + \frac{|\sharp_\Delta(i,j)|}{\min\{d_i,d_j\}} + \frac{(\gamma_{max})^{-1}}{\max\{d_i,d_j\}}(|\sharp_\square^i| + |\sharp_\square^j|), \quad (3)$$

*where the last term is set to be zero if* $|\sharp_\square^i|$ *(and hence* $|\sharp_\square^j|$*) is zero. In particular* $\mathrm{Ric}(i,j) > -2$.

The curvature is negative when $i \sim j$ behaves as a *bridge* between $S_1(i)$ and $S_1(j)$, while it is positive when $S_1(i)$ and $S_1(j)$ stay connected after removing $i \sim j$. We refer to Ric as *Balanced Forman curvature*. We can relate the Balanced Forman curvature to the Jacobian of hidden features, while also extending many results valid for the Ollivier curvature $\kappa(i,j)$ thanks to our next theorem.

**Theorem 2.** *Given an unweighted graph $G$, for any edge $i \sim j$ we have* $\kappa(i,j) \geq \mathrm{Ric}(i,j)$.

Theorem 2 generalizes Jost & Liu (2014, Theorem 3) (see Appendix). We also note that the computational complexity for $\kappa$ scales as $O(|E|d_{\max}^3)$, while for our Ric we have $O(|E|d_{\max}^2)$, with $d_{\max}$ the maximal degree. From Theorem 2 and Paeng (2012), we find:

**Corollary 3.** *If* $\mathrm{Ric}(i,j) \geq k > 0$ *for any edge $i \sim j$, then there exists a polynomial $P$ such that*

$$|B_r(i)| \leq P(r), \quad \forall i \in V.$$

| Graph $G$ | | $\mathrm{Ric}_G$ |
|---|---|---|
| | $C_3$ | $\frac{3}{2}$ |
| Cycles | $C_4$ | $1$ |
| | $C_{n \geq 5}$ | $0$ |
| Complete $K_n$ | | $\frac{n}{n-1}$ |
| Grid $G_n$ | | $0$ |
| Tree $T_r$ | | $\frac{4}{r+1} - 2$ |

Table 1: Examples of the Balanced Forman curvature.

Therefore, when the curvature is positive everywhere, the bottleneck effect should not play a crucial role as the receptive field of each node will be polynomial in the hop-distance. This limit case shows how the curvature determines whether a learning task on a graph would suffer from over-squashing.

**Curvature and over-squashing.** Thanks to our new combinatorial curvature and the sensitivity analysis in Lemma 1, we are able to relate local curvature properties to the Jacobian of the node representations. This leads to one of the main results of this paper: **negatively curved edges are those causing the graph bottleneck and thus leading to the over-squashing phenomenon:**

**Theorem 4.** *Consider a MPNN as in equation 1. Let $i \sim j$ with $d_i \leq d_j$ and assume that:*

*(i) $|\nabla \phi_\ell| \leq \alpha$ and $|\nabla \psi_\ell| \leq \beta$ for each $0 \leq \ell \leq L-1$, with $L \geq 2$ the depth of the MPNN.*

*(ii) There exists $\delta$ s.t. $0 < \delta < (\max\{d_i, d_j\})^{-\frac{1}{2}}$, $\delta < \gamma_{max}^{-1}$, and $\mathrm{Ric}(i,j) \leq -2 + \delta$.*

*Then there exists $Q_j \subset S_2(i)$ satisfying $|Q_j| > \delta^{-1}$ and for $0 \leq \ell_0 \leq L-2$ we have*

$$\frac{1}{|Q_j|} \sum_{k \in Q_j} \left| \frac{\partial h_k^{(\ell_0+2)}}{\partial h_i^{(\ell_0)}} \right| < (\alpha\beta)^2 \delta^{\frac{1}{4}}. \tag{4}$$

Condition (i) is always satisfied and allows us to control the message passing functions. The requirement (ii) instead means that the curvature of $(i,j)$ is negative enough when compared to the degrees of $i$ and $j$ (recall that $\mathrm{Ric}(i,j) > -2$). The further condition on $\gamma_{\max}$ is to avoid pathological cases where we have a large number of degenerate 4-cycles passing through the same three nodes.

To understand the conclusions in Theorem 4, let us fix $\ell_0 = 0$. The equation 4 shows that negatively curved edges are the ones causing bottlenecks, interpreted as how the graph topology prevents a representation $h_k^{(2)}$ to be affected by *non*-adjacent features $h_i^{(0)} = x_i$. Theorem 4 implies that if we have a negatively curved edge as in (ii), then there exist a large number of nodes $k$ such that GNNs - on average - struggle to propagate messages from $i$ to $k$ in two layers despite these nodes $k$ being at distance 2 from $i$. In this case the over-squashing occurs as measured by the Jacobian in equation 4 and hence the propagation of information suffers. If the task at hand has long-range dependencies, then the over-squashing caused by the negatively curved edges may compromise the performance.

**Bottleneck via Cheeger constant.** We now relate the previous discussion about bottlenecks and curvature to spectral properties of the graph. In particular, since the spectral gap of a graph can be interpreted as a topological obstruction to the graph being partitioned into two communities, we argue below that this quantity is related to the graph bottleneck and should hence be controllable by the curvature. We start with an intuitive explanation: suppose we are given a graph $G$ with two communities separated by few edges. In this case, we see that the graph can be easily disconnected. This property is encoded in the classical notion of the *Cheeger constant* (Chung & Graham, 1997)

$$h_G := \min_{S \subset V} h_S, \quad h_S := \min_{S \subset V} \frac{|\partial S|}{\min\{\mathrm{vol}(S), \mathrm{vol}(V \setminus S)\}} \tag{5}$$

where $\partial S = \{(i,j) : i \in S, \ j \in V \setminus S\}$ and $\mathrm{vol}(S) = \sum_{i \in S} d_i$. The main result about the Cheeger constant is the *Cheeger inequality* (Cheeger, 2015; Chung & Graham, 1997):

$$2h_G \geq \lambda_1 \geq \frac{h_G^2}{2} \tag{6}$$

where $\lambda_1$ is the first non-zero eigenvalue of the normalized graph Laplacian, often referred to as the *spectral gap*. A graph with two tightly connected communities ($S$ and $V \setminus S := \bar{S}$) and few inter-community edges has a small Cheeger constant $h_G$. For nodes in different communities to interact with each other, all messages need to go through the same few *bridges* hence leading to the over-squashing of information (a similar intuition was explored in Alon & Yahav (2021)). Therefore, $h_G$ can be interpreted as a rough measure of graph 'bottleneckedness', in the sense that the smaller its value, the more likely the over-squashing is to occur across inter-community edges. Since Theorem 4 implies that negatively curved edges induce the bottleneck, we expect a relationship between $h_G$ and the curvature of the graph. The next proposition follows from Theorem 2 and Lin et al. (2011):

**Proposition 5.** *If $\mathrm{Ric}(i,j) \geq k > 0$ for all $i \sim j$, then $\lambda_1/2 \geq h_G \geq \frac{k}{2}$.*

Therefore, a positive lower bound on the curvature gives us a control on $h_G$ and hence on the spectral gap of the graph. In the next section, we show that diffusion-based graph-rewiring methods might fail to significantly alter $h_G$ and hence correct the graph bottleneck potentially induced by inter-community edges. This will lead us to propose an alternative curvature-based graph rewiring.

## 4 CURVATURE-BASED REWIRING METHODS

The traditional paradigm of message passing graph neural networks assumes that messages are propagated on the input graph (Gilmer et al., 2017). More recently, there is a trend to decouple the input graph from the graph used for information propagation. This can take the form of graph subsampling or resampling to deal with scalability (Hamilton et al., 2017) or topological noise (Zhang et al., 2019), using larger motif-based (Monti et al., 2018) or multi-hop filters (Rossi et al., 2020), or changing the graph either as a preprocessing step (Klicpera et al., 2019; Alon & Yahav, 2021) or adaptively for the downstream task (Wang et al., 2019; Kazi et al., 2020). Such methods are often generically referred to as *graph rewiring*.

In the context of this paper, we assume that graph rewiring attempts to produce a new graph $G' = (V, E')$ with a different edge structure that reduces the bottleneck and hence potentially alleviates the over-squashing of information. We propose a method that leverages the graph curvature to guide the rewiring steps in a surgical way by modifying the negatively-curved edges, so to decrease the bottleneck without significantly compromising the statistical properties of the input graph. We also rigorously show that a random-walk based rewiring method might generally fail to obtain an edge set $E'$ with a significant improvement in its bottleneckedness as measured by the Cheeger constant.

**Curvature-based graph rewiring.** Since according to Theorem 4 negatively curved edges induce a bottleneck and are hence responsible for over-squashing, a curvature-based rewiring method should attempt to alleviate a graph's strongly-negatively curved edges. To this end we implement a simple rewiring method called Stochastic Discrete Ricci Flow (SDRF), described in Algorithm 1.

---

**Algorithm 1:** Stochastic Discrete Ricci Flow (SDRF)

---

**Input:** graph $G$, temperature $\tau > 0$, max number of iterations, optional Ric upper-bound $C^+$
**Repeat**
    1) For edge $i \sim j$ with minimal Ricci curvature $\text{Ric}(i, j)$:
        Calculate vector $\boldsymbol{x}$ where $x_{kl} = \text{Ric}_{kl}(i, j) - \text{Ric}(i, j)$, the improvement to $\text{Ric}(i, j)$
        from adding edge $k \sim l$ where $k \in B_1(i)$, $l \in B_1(j)$;
        Sample index $k, l$ with probability softmax$(\tau \boldsymbol{x})_{kl}$ and add edge $k \sim l$ to $G$.
    2) Remove edge $i \sim j$ with maximal Ricci curvature $\text{Ric}(i, j)$ if $\text{Ric}(i, j) > C^+$.
**Until** convergence, or max iterations reached;

---

At each iteration this preprocessing step adds an edge to 'support' the graph's most negatively curved edge, and then removes the most positively curved edge. The requirement on the added edge $k \sim l$ that $k \in B_1(i)$ and $l \in B_1(j)$ ensures that we're adding either an extra 3- or 4-cycle around the negative edge $i \sim j$ so that this is a local modification. The graph edit distance between the original and preprocessed graph is bounded above by $2 \times$ the max number of iterations. The temperature $\tau$ determines how stochastic the edge addition is, with $\tau = \infty$ being fully deterministic (the best edge is always added). At each step we remove the edge with most positive curvature to balance the distributions of curvature and node degrees. We use Balanced Forman curvature as in equation 3 for $\text{Ric}(i, j)$. $C^+$ can be chosen to stop the method skewing the curvature distribution negative, including $C^+ = \infty$ to not remove any edges. The method is inspired by the continuous (backwards) Ricci flow with the aim of homogenizing edge curvatures. This is different from more direct extensions of Ricci flow on graphs where it becomes increasingly expensive to propagate messages across negatively curved edges (as in other applications such as Ni et al. (2019)). An example alongside its continuous analogue can be seen in Figure 1.

**Can random-walk based rewiring address bottlenecks?** A good way of understanding the effectiveness of SDRF in reducing the graph bottleneck is through comparison with random-walk based rewiring strategies. Recall that, as argued in Section 3, the Cheeger constant $h_G$ of a graph constitutes a rough measure of its bottleneckedness as induced by the inter-community edges (a small $h_G$ is indicative of a bottleneck). Suppose we are given a graph $G$ with a small $h_G$ and wish to rewire it into a graph $G'$ with a significantly improved Cheeger constant in order to reduce the inter-community bottleneck. A random-walk based rewiring method such as DIGL (Klicpera et al., 2019) acts by smoothing out the graph adjacency and hence tends to promote connections among nodes at short

*diffusion distance* (Coifman & Lafon, 2006). Accordingly, such a rewiring method might fail to correct structural features like the bottleneck, which is instead more prominent for nodes that are at long diffusion distance.[3] To emphasize this point, we consider a classic example: given $\alpha \in (0,1)$, the Personalized Page Rank (PPR) matrix is defined by (Brin & Page, 1998) as

$$R_\alpha := \sum_{k=0}^{\infty} \theta_k^{PPR}(D^{-1}A)^k = \alpha \sum_{k=0}^{\infty} \left((1-\alpha)(D^{-1}A)\right)^k.$$

Assume that we rewire the graph using $R_\alpha$ as in Klicpera et al. (2019) with the PPR kernel, meaning that we replace the given adjacency $A$ with $R_\alpha$. Since $R_\alpha$ is stochastic, the new Cheeger constant of the rewired graph can be computed as

$$h_{S,\alpha} = \frac{|\partial S|_\alpha}{\text{vol}_\alpha(S)} \equiv \frac{1}{|S|} \sum_{i \in S} \sum_{j \in \bar{S}} (R_\alpha)_{ij}.$$

By applying (Chung, 2007, Lemma 5), we show that we cannot improve the Cheeger constant (and hence the bottleneck) arbitrarily well (in contrast to a curvature-based approach). We refer to Proposition 17 and Remark 18 in Appendix E for results that are more tailored to the actual strategy adopted in Klicpera et al. (2019) where we also take into account the effect of the sparsification.

**Theorem 6.** *Let $S \subset V$ with $vol(S) \leq vol(G)/2$. Then $h_{S,\alpha} \leq \left(\frac{1-\alpha}{\alpha}\right) \frac{d_{\text{avg}}(S)}{d_{\min}(S)} h_S$, where $d_{\text{avg}}(S)$ and $d_{\min}(S)$ are the average and minimum degree on S, respectively.*

The property that the new Cheeger constant is directly controlled by the old one stems from the fact that a random-walk approach like in Klicpera et al. (2019) is meant to act more relevantly on intra-community edges rather than inter-community edges because it prioritizes short diffusion distance nodes. This is also why this method performs well on high-homophily datasets, as discussed below. In particular, for a fixed $\alpha \in (0,1)$, the bound in Theorem 6 can be very small. As a specific example, consider two complete graphs $K_n$ joined by one bridge. Then $h_G = (n(n-1)+1)^{-1}$, which means that the bound on the right hand side is $O(n^{-2})$.

Theorem 6 implies that a diffusion approach such as DIGL might fail to yield a new edge set $E'$ with a sufficiently improved bottleneck. By contrast, from Theorem 4 and Proposition 5, we deduce that a curvature-based rewiring method such as SDRF properly addresses the edges that cause the bottleneck.

**Graph structure preservation.** Although a graph-rewiring approach aims at providing a new edge set $E'$ potentially more beneficial for the given learning task, it is still desirable to control how far $E'$ is from $E$. In this regard, we note that a curvature-based rewiring is surgical in nature and hence more likely to preserve the structure of the input graph better than a random-walk based approach. Consider, for example, that we are given $\rho > 0$ and wish to rewire the graph such that the new edge set $E'$ is within graph-edit distance $\rho$ from the original $E$. Theorem 4 tells us how to do the rewiring under such constraints in order to best address the over-squashing: the topological modifications need to be localized *around the most negatively-curved edges*. We can do this with SDRF, with the maximum number of iterations set to $\rho/2$.

Secondly, we also point out that $\text{Ric}(i,j) < -2 + \delta$ implies that $\min\{d_i, d_j\} > 2/\delta$. Therefore, if we mostly modify the edge set at those nodes $i, j$ joined by an edge with large negative curvature, then we are perturbing nodes with high degree where such a change is relatively insignificant, and thus overall statistical properties of the rewired graph such as degree distribution are likely to be better preserved. Moreover, graph convolutional networks tend to be more stable to perturbations of high degree nodes (Zügner et al., 2020; Kenlay et al., 2021), making curvature-based rewiring more suitable for the downstream learning tasks with popular GNN architectures.

**Homophily and bottleneck.** As a final remark, note that the graph rewiring techniques considered in this paper (both DIGL and SDRF) are based purely on the topological structure of the graph and completely agnostic to the node features and to whether the dataset is *homophilic* (adjacent nodes have same labels) or *heterophilic*. Nonetheless, the different nature of these rewiring methods allows us to draw a few broad conclusions about their suitability in each of these settings. A random-walk

---

[3]We refer to the right hand side of equation 2 where the power of the normalized augmented adjacency is measuring the number of walks of distance $r$ from $i$ to $s$.

| $\mathcal{H}(G)$ | Cornell 0.11 | Texas 0.06 | Wisconsin 0.16 | Chameleon 0.25 | Squirrel 0.22 | Actor 0.24 | Cora 0.83 | Citeseer 0.71 | Pubmed 0.79 |
|---|---|---|---|---|---|---|---|---|---|
| None | $52.69 \pm 0.21$ | $61.19 \pm 0.49$ | $54.60 \pm 0.86$ | $41.80 \pm 0.41$ | $39.83 \pm 0.14$ | $28.70 \pm 0.09$ | $81.89 \pm 0.79$ | $72.31 \pm 0.17$ | $78.16 \pm 0.23$ |
| Undirected | $53.20 \pm 0.53$ | $63.38 \pm 0.87$ | $51.37 \pm 1.15$ | $42.63 \pm 0.30$ | $40.77 \pm 0.16$ | $28.10 \pm 0.11$ | - | - | - |
| +FA | $\mathbf{58.29 \pm 0.49}$ | $\mathbf{64.82 \pm 0.29}$ | $55.48 \pm 0.62$ | $42.33 \pm 0.17$ | $40.74 \pm 0.13$ | $28.68 \pm 0.16$ | $81.65 \pm 0.18$ | $70.47 \pm 0.18$ | $\mathbf{79.48 \pm 0.12}$ |
| DIGL (PPR) | $58.26 \pm 0.50$ | $62.03 \pm 0.43$ | $49.53 \pm 0.27$ | $42.02 \pm 0.13$ | $34.38 \pm 0.11$ | $\mathbf{30.79 \pm 0.10}$ | $\mathbf{83.21 \pm 0.27}$ | $\mathbf{73.29 \pm 0.17}$ | $78.84 \pm 0.08$ |
| DIGL + Undirected | $\mathbf{59.54 \pm 0.64}$ | $63.54 \pm 0.38$ | $52.23 \pm 0.54$ | $42.68 \pm 0.12$ | $33.36 \pm 0.21$ | $29.71 \pm 0.11$ | - | - | - |
| SDRF | $54.60 \pm 0.39$ | $64.46 \pm 0.38$ | $\mathbf{55.51 \pm 0.27}$ | $\mathbf{43.75 \pm 0.31}$ | $40.97 \pm 0.14$ | $29.70 \pm 0.13$ | $\mathbf{82.76 \pm 0.23}$ | $72.58 \pm 0.20$ | $79.10 \pm 0.11$ |
| SDRF + Undirected | $57.54 \pm 0.34$ | $\mathbf{70.35 \pm 0.60}$ | $\mathbf{61.55 \pm 0.86}$ | $\mathbf{44.46 \pm 0.17}$ | $\mathbf{41.47 \pm 0.21}$ | $\mathbf{29.85 \pm 0.07}$ | - | - | - |

Table 2: Experimental results on common node classification benchmarks. Top two in bold.

approach such as DIGL tends to improve the connectivity among nodes that are at short diffusion distance; since for a high-homophily dataset these nodes often share the same label, a rewiring method like DIGL is likely to act as *graph denoising* and yield improved performance. On the other hand, for datasets with low homophily, nodes at short diffusion distance are more likely to belong to different label classes, meaning that a diffusion-based rewiring might inject noise and hence compromise performance as also noted in Klicpera et al. (2019). Conversely, on a low-homophily dataset, a curvature-based approach as SDRF modifies the edge set mainly around the most negatively curved edges, meaning that it decreases the bottleneck without significantly increasing the connectivity among nodes with different labels. In fact, long-range dependencies are often more relevant in low-homophily settings, where nodes sharing the same labels are in general not neighbors. This observation is largely confirmed by experimental results reported in the next section.

## 5 EXPERIMENTAL RESULTS

**Experiment setup.** To demonstrate the theoretical results in this paper we ran a suite of semi-supervised node classification tasks comparing our curvature-based rewiring method SDRF to DIGL from Klicpera et al. (2019) (GDC with the PPR kernel) and the +FA method from Alon & Yahav (2021), where the last layer of the GNN is made fully connected. We evaluate the methods on nine datasets: Cornell, Texas and Wisconsin from the WebKB dataset[4]; Chameleon and Squirrel (Rozemberczki et al., 2021) along with Actor (Tang et al., 2009); and Cora (McCallum et al., 2000), Citeseer (Sen et al., 2008) and Pubmed (Namata et al., 2012). Statistics for these datasets can be found in Appendix F.1. Our base model is a GCN (Kipf & Welling, 2017). Following Shchur et al. (2018) and Klicpera et al. (2019) we optimized hyperparameters for all dataset-preprocessing combinations separately by random search over 100 data splits. Results are reported as average accuracies on a test set used once with 95% confidence intervals calculated by bootstrapping. We compared the performance on graphs with no preprocessing, making the graph undirected, +FA, DIGL, SDRF, and the given combinations. For DIGL + Undirected we symmetrized the diffusion matrix as in Klicpera et al. (2019), and for SDRF + Undirected we made the graph undirected before applying SDRF. For more details on the experiments and datasets see Appendix F, and for the hyperparameters used for each model and preprocessing see Appendix F.4.

**Node classification results.** Table 2 shows the results of the experiments. As well as reporting results we give a measure of homophily $\mathcal{H}(G)$ proposed by Pei et al. (2019) (restated in Appendix F, equation 21), by which we can see our experiment set is diverse with respect to homophily. We see that SDRF improves upon the baseline in all cases, and that the largest improvements are seen on the low-homophily datasets. We also see that SDRF matches or outperforms DIGL and +FA on most datasets, supporting our argument that curvature-based rewiring is a viable candidate for improving GNN performance.

**Graph topology change.** Furthermore, SDRF preserves the graph topology to a far greater extent than DIGL due to its surgical nature. Table 3 shows the number of edges added / removed by the two preprocessings on each dataset as a percentage of the original number of edges. We see that for optimal performance DIGL makes the graph much denser, which significantly affects the node degrees and may negatively impact the time and space complexity of the downstream GNN, which typically are $O(E)$. In comparison, SDRF adds and removes a similar number of edges and approximately preserves the degree distribution. The effect on the full degree distribution for three

---

[4]http://www.cs.cmu.edu/afs/cs.cmu.edu/project/theo-11/www/wwkb/

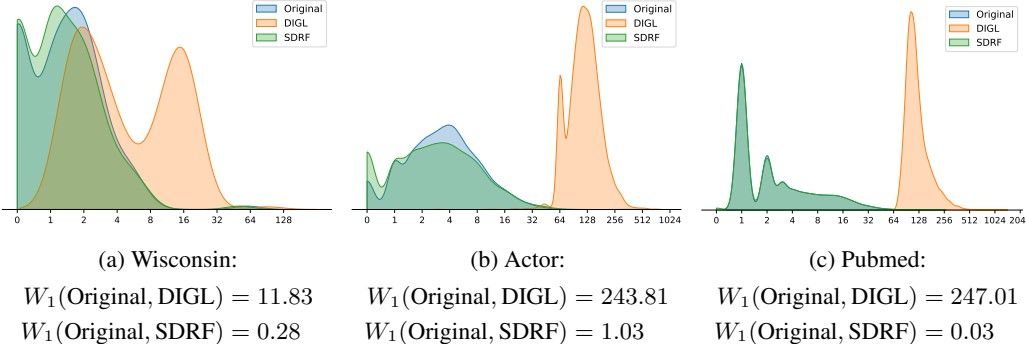

(a) Wisconsin:

$W_1(\text{Original}, \text{DIGL}) = 11.83$

$W_1(\text{Original}, \text{SDRF}) = 0.28$

(b) Actor:

$W_1(\text{Original}, \text{DIGL}) = 243.81$

$W_1(\text{Original}, \text{SDRF}) = 1.03$

(c) Pubmed:

$W_1(\text{Original}, \text{DIGL}) = 247.01$

$W_1(\text{Original}, \text{SDRF}) = 0.03$

Figure 4: Comparing the degree distribution of the original graphs to the preprocessed version. The x-axis is node degree in $\log_2$ scale, and the plots are a kernel density estimate of the degree distribution. In the captions we see the Wasserstein distance $W_1$ between the original and preprocessed graphs.

datasets is shown in Figure 4. We see that the degree distribution following SDRF is close to, or in some cases indistinguishable from, the original distribution, whereas DIGL has a more noticeable effect. We also compute the Wasserstein distance between the degree distributions, denoted by $W_1$ in the figure captions, to numerically confirm this observation. For this analysis extended to all nine datasets see Appendix F.2.

This difference is also visually evident from Figure 6 in Appendix F.3, where we again observe that SDRF largely preserves the topology of the Cornell graph and that the curvature (encoded by the edge color) is homogenized across the graph. We also see that the entries of the trained

|  | DIGL | SDRF |
|---|---|---|
| Cornell | 351.1% / 0.0% | 7.8% / 33.3% |
| Texas | 483.3% / 0.0% | 2.4% / 10.4% |
| Wisconsin | 300.6% / 0.0% | 1.4% / 7.5% |
| Chameleon | 336.1% / 11.8% | 6.4% / 6.4% |
| Squirrel | 228.8% / 1.9% | 0.7% / 0.7% |
| Actor | 2444.0% / 2.3% | 5.4% / 9.9% |
| Cora | 3038.0% / 0.5% | 1.0% / 1.0% |
| Citeseer | 2568.3% / 0.0% | 1.1% / 1.1% |
| Pubmed | 2747.1% / 0.1% | 0.2% / 0.2% |

Table 3: % edges added / removed by method.

network's Jacobian between a node's prediction and the features of its 2-hop neighbors (encoded as node colors) are increased over the graph, which we may attribute to both DIGL and SDRF's ability to alleviate the upper bound presented in Theorem 4 and thus reduce over-squashing.

## 6 CONCLUSION

In this paper, we studied the graph bottleneck and the over-squashing phenomena limiting the performance of message passing graph neural networks from a geometric perspective. We started with a Jacobian approach to determine how the over-squashing phenomenon is dictated by the graph topology as in equation 2. We then investigated further how the topology induces the bottleneck and hence causes over-squashing. We introduced a new notion of edge-based Ricci curvature called Balanced Forman curvature, relating it to the classical Ollivier curvature (Theorem 2). We then proved in Theorem 4 that negatively-curved edges are responsible for over-squashing, calling for a possibility of curvature-based rewiring of the graph in order to improve its bottleneckedness. We show one such possibility (Algorithm 1), inspired by the classical Ricci flow and comment on the advantages of surgical method such as this. We show both theoretically and experimentally that the proposed method can be advantageous compared to a diffusion-based rewiring approach, opening the door for curvature-based rewiring methods for improving GNN performance going forward.

**Limitations and future directions.** Our paper establishes a geometric perspective on the graph bottleneck and over-squashing, providing new tools to study and cope with these phenomena. One limitation of our work is that the theoretical results presented here do not currently extend to multi-graphs. In addition, the current methodology is agnostic to information beyond the graph topology, such as node features. In future works, we will develop a notion of the curvature and the corresponding rewiring method that can take into account such information.

**Acknowledgements.** This research was supported in part by the EPSRC CDT in Modern Statistics and Statistical Machine Learning (EP/S023151/1) and the ERC Consolidator Grant No. 724228 (LEMAN). X.D. gratefully acknowledges support from the Oxford-Man Institute of Quantitative Finance and the EPSRC (EP/T023333/1).

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

APPENDIX

The Appendix is structured as follows:

(i) In Appendix A we prove Lemma 1 and a side-result about the role of self-loops. We also summarize how to extend the analysis in Lemma 1 to message passing models with sum aggregations (not average), meaning those architectures where we do not normalize the adjacency matrix.

(ii) In Appendix B we introduce and describe different quantities we use to characterize the neighbourhood of a given edge $i \sim j$. These objects are all essential to studying the new notion of balanced Forman curvature. The focus is on how we can distinguish 4-cycles in a computationally tractable way without losing too much accuracy.

(iii) In Appendix C we provide a brief literature review about existing curvature candidates. In particular, we report the definitions of (modified) Ollivier curvature and Forman curvature.

(iv) In Appendix D we prove the statements in Section 3, i.e. Theorem 2, Corollary 3, Theorem 4 and Proposition 5. We also comment on the role of the assumptions and compare the bound in Theorem 2 with the existing literature. Finally, we relate the classical notion of betweenness centrality to the over-squashing effect and the negatively curved edges in a graph.

(v) In Appendix E we prove the results in Section 4, namely Theorem 6 and an analogous result.

(vi) In Appendix F we describe more fully the experiments from Section 5, including a full analysis on degree distribution (Appendix F.2) and the hyperparameters used in our experiments (Appendix F.4).

(vii) In Appendix G we comment on hardware specifications.

## A    PROOFS OF RESULTS IN SECTION 2

**Lemma 1.** *Assume an MPNN as in equation 1. Let $i, s \in V$ with $s \in S_{r+1}(i)$. If $|\nabla \phi_\ell| \leq \alpha$ and $|\nabla \psi_\ell| \leq \beta$ for $0 \leq \ell \leq r$, then*

$$\left| \frac{\partial h_i^{(r+1)}}{\partial x_s} \right| \leq (\alpha\beta)^{r+1}(\hat{A}^{r+1})_{is}. \tag{2}$$

*Proof.* Let $i \in V$ and $s \in S_{r+1}(i)$. We recall that to ease the notations we assume that node features and hidden representations are scalar. The proof in the more general higher-dimensional case follows without any modification. We compute

$$\frac{\partial h_i^{(r+1)}}{\partial x_s} = \partial_1 \phi_r(\ldots) \partial_{x_s} h_i^{(r)}$$

$$+ \partial_2 \phi_r(\ldots) \sum_{j_r=1}^n \hat{a}_{ij_r} \left( \partial_1 \psi_r(h_i^{(r)}, h_{j_r}^{(r)}) \partial_{x_s} h_i^{(r)} + \partial_2 \psi_r(h_i^{(r)}, h_{j_r}^{(r)}) \partial_{x_s} h_{j_r}^{(r)} \right).$$

We can iterate the computation above and see that the right hand side can be expanded as

$$\frac{\partial h_i^{(r+1)}}{\partial x_s} = \sum_{j_r,\ldots,j_0} \sum_{k_r \in \{i,j_r\}} \cdots \sum_{k_1 \in \{i,j_r,\ldots,j_1\}} \hat{a}_{ij_r} \hat{a}_{k_r j_{r-1}} \ldots \hat{a}_{k_1 j_0} Z_{ij_r k_r j_{r-1}\ldots k_1 j_0}(X) \partial_{x_s} h_{j_0}^{(0)},$$

for some functions $Z_{ij_r\ldots k_1 j_0}$ of the input features obtained as products of $r+1$ partial derivatives of the maps $\phi_\ell$ and $r+1$ partial derivatives of the maps $\psi_\ell$. Since $H^{(0)} = X$, we have

$$\partial_{x_s} h_{j_0}^{(0)} = \delta_{j_0 s}$$

meaning that the previous sum becomes

$$\frac{\partial h_i^{(r+1)}}{\partial x_s} = \sum_{j_r,\ldots,j_1} \sum_{k_r \in \{i,j_r\}} \cdots \sum_{k_1 \in \{i,j_r,\ldots,j_1\}} \hat{a}_{ij_r}\hat{a}_{k_r j_{r-1}}\ldots\hat{a}_{k_1 s}Z_{ij_r k_r j_{r-1}\ldots k_1 s}(X).$$

Since $d_G(i,s) = r + 1$, the only non-vanishing terms in the sum above are the minimal walks from $i$ to $s$. In fact, if there existed a different choice of coefficients yielding a non-zero term then we would find a walk joining $i$ to $s$ of length lesser than $r + 1$, which is in contradiction with the definition of geodesic distance. Since $Z_{i\ldots s}(X)$ is a product of $r + 1$-partial derivatives of the aggregation and update maps and by assumption their gradients are bounded by $\alpha$ and $\beta$ respectively, we conclude that

$$\left|\frac{\partial h_i^{(r+1)}}{\partial x_s}\right| \leq (\alpha\beta)^{r+1} \sum_{j_r,\ldots,j_1} \hat{a}_{ij_r}\hat{a}_{j_r j_{r-1}}\ldots\hat{a}_{j_1 s} = \alpha^{r+1}(\hat{A}^{r+1})_{is}$$

which completes the proof of the Lemma. □

As a byproduct of this analysis, we can also provide a rigorous motivation for the role of self-loops in GNNs (see Appendix for details):

**Corollary 7.** *If $h_i^{(\ell+1)} = \sum_{j\sim i} \psi_\ell(h_j^{(\ell)})$, then $h_i^{(\ell+1)}$ only depends on nodes that can be reached via walks of length **exactly** $\ell + 1$. By adding self-loops, the GNN also takes into account nodes that can be reached via walks of length $r \leq \ell + 1$.*

*Proof.* If $h_i^{(\ell+1)} = \sum_{j\sim i} \psi_\ell(h_j^{(\ell)})$ for each $\ell \in [0, L-1]$, then we can argue as in the proof of Lemma 1 and find

$$\frac{\partial h_i^{(\ell+1)}}{\partial x_s} = \sum_{j_\ell,\ldots,j_1} a_{ij_\ell}\ldots a_{j_1 s}\psi_\ell'(h_{j_\ell}^{(\ell)})\cdots\psi_0'(x_s).$$

The combinatorial coefficient $a_{ij_\ell}\ldots a_{j_1 s}$ is non-zero iff there exists a walk from $i$ to $s$ of length exactly $\ell+1$, since we are not taking into account the contribution coming from self-loops. Conversely, if each term $a_{ij_\ell}$ was replaced by $\hat{a}_{ij_\ell}$ then we would find that $\hat{a}_{ij_\ell}\ldots\hat{a}_{j_1 s}$ is non-zero iff there exists a walk from $i$ to $s$ of length *at most* $r + 1$, since the diagonal entries are now positive. □

**Remark 8.** *As a specific instance of Corollary 7, we note that if we do not include self-loops in the adjacency matrix, then the output of a 2-layer simplified graph neural network at node $i$ is independent of the features of neighbours $k$ that do not form a triangle with $i$. Once again here the dependence is precisely measured via the Jacobian of the hidden features with respect to the input features.*

**Remark 9.** *We note that the role of self-loops has also implicitly been noted in Xu et al. (2018) where the analysis of the Jacobian of node representations on the graph augmented with self-loops has been related to lazy random-walks.*

**GNNs with different aggregations** We note that similar conclusions extend to message passing architectures where the aggregations are sums and not averages meaning that we take the augmented adjacency without normalizing by the degree matrices. Consistently with Lemma 1, we restrict to the setting where features and node representations at each layer are scalars to make the discussion simpler. In line with the Xu et al. (2018) we consider a GNN-model of the form

$$h_i^{(\ell+1)} = \text{ReLU}\left(\sum_{j\in\tilde{\mathcal{N}}_i} h_j^{(\ell)}w_\ell\right).$$

Note that the augmented neighbourhood $\tilde{\mathcal{N}}_i$ is defined as $\mathcal{N}_i \cup \{i\}$. Differently from the setting of Theorem 1 in Xu et al. (2018), the aggregation here is not an average but a simple sum. Let us now take nodes $i$ and $s$ such that $s \in S_{r+1}(i)$ as in the statement of Lemma 1. In this case, instead of simply considering the quantity $|\partial h_i^{(r+1)}/\partial x_s|$, we normalize the Jacobian entries - obtaining what

is referred to as *influence score* in Xu et al. (2018):

$$\mathcal{J}_{r+1}(i,s) := \frac{\left| \frac{\partial h_i^{(r+1)}}{\partial x_s} \right|}{\sum_k \left| \frac{\partial h_i^{(r+1)}}{\partial x_k} \right|}$$

This of course represents now a relative importance of feature $x_s$ on the representation of node $i$ at layer $r+1$. If - similarly to Theorem 1 in Xu et al. (2018) - we assume that all paths in the computational graph of the model are activated with the same probability, then we obtain that on average

$$\mathcal{J}_{r+1}(i,s) = \frac{\tilde{A}_{is}^{r+1}}{\sum_k \tilde{A}_{ik}^{r+1}} \leq \frac{\tilde{A}_{is}^{r+1}}{\text{Vol}(B_{r+1}(i))},$$

where $\tilde{A} = A + I$ and $\text{vol}(S) = \sum_{j \in S} d_j$. In particular, we again find that if we have a tree structure, then the right hand side decays exponentially as $2^{-(r+1)}$.

## B PRELIMINARY ANALYSIS OF AN EDGE-NEIGHBORHOOD

Given an edge $i \sim j$, we introduce the sets below:

(i) $\sharp_\triangle(i,j) := S_1(i) \cap S_1(j)$, the number of triangles based at the edge $i \sim j$.

(ii) $\sharp_\square^i(i,j) := \{k \in S_1(i) \setminus S_1(j), k \neq j : \exists w \in (S_1(k) \cap S_1(j)) \setminus S_1(i)\}$, the number of nodes $k \in S_1(i)$ forming a 4-cycle based at $i \sim j$ *without* diagonals inside.

(iii) $Q_i(j) := S_1(i) \setminus (\{j\} \cup \sharp_\triangle(i,j) \cup \sharp_\square^i(i,j))$, simply the complement of the neighbours of $i$ with respect to the sets introduced in (i) and (ii) once we also exclude $j$.

In the following we simply write $\sharp_\triangle$, $\sharp_\square^i$ and $Q_i$ when the edge $i \sim j$ is clear from the context.

**4-cycle contributions.** In general the sets $\sharp_\square^i$ and $\sharp_\square^j$ may differ. This may occur when there exists a node $k$ belonging to $\sharp_\square^i$ that admits multiple solutions $w$ as in the definition of $\sharp_\square^i$. This feature needs to be taken into account when comparing Ollivier's Ricci curvature to the new notion we present below. We first introduce the following class to ease the notations.

**Definition 2.** *For any simple, undirected graph $G = (V, E)$, if $U \subset V$, then we set*

$$\mathcal{D}(U) := \{\varphi : U \to V, \ |U| = |\varphi(U)|, \ (z, \varphi(z)) \in E, \ \forall z \in U\}.$$

We note that any $\varphi \in \mathcal{D}(U)$ is injective.
We may now define a quantity which measures the maximal number of 1-1 pairings that can be performed from $\sharp_\square^i$ to $\sharp_\square^j$.

**Definition 3.** *For any edge $i \sim j$ we let*

$$\sharp_\square^m(i,j) := \max \left\{ |U| : \ U \subset \sharp_\square^i, \ \exists \varphi : U \to \sharp_\square^j, \ \varphi \in \mathcal{D}(U) \right\}.$$

We often simply write $\sharp_\square^m$. While the quantity $\sharp_\square^m$ plays a role in the derivation of the Ollivier curvature of $i \sim j$ it is not computationally-friendly, as to determine $\sharp_\square^m$ we need to identify and distinguish all possible 4-cycles based at $i \sim j$ and then choose a maximal pairing map. Accordingly, we consider a looser term which is easier to compute:

**Definition 4.** *For any pair of adjacent nodes $i \sim j$ we define*

$$\gamma_{max}(i,j) := \max \left\{ \max_{k \in \sharp_\square^i} \{(A_k \cdot (A_j - A_i \odot A_j)) - 1\}, \max_{w \in \sharp_\square^j} \{(A_w \cdot (A_i - A_j \odot A_i)) - 1\} \right\},$$

*where $A_s$ denotes the $s$-th row of the adjacency matrix. We usually simply write $\gamma_{max}$.*

**Remark 10.** *We note that given $k \in S_1(i) \setminus S_1(j)$ the term $(A_k \cdot (A_j - A_i \odot A_j)) - 1$ yields the number of nodes $w$ forming a 4-cycle of the form $i \sim k \sim w \sim j \sim i$ with no diagonals inside. The value $\gamma_{max}$ measures the maximal degeneracy of edges forming 4-cycles, meaning that it is equal to 1 iff for each $k \in \sharp_\Box^i$ there exists a unique node $w \in \sharp_\Box^j$ such that $i \sim k \sim w \sim j \sim i$ is a 4-cycle.*

We now end the discussion about 4-cycle contributions by proving the following inequality, which allows us to avoid to compute directly the term $\sharp_\Box^m$ up to giving up some accuracy.

**Lemma 11.** *For any edge $i \sim j$ we have*

$$|\sharp_\Box^m| \geq \frac{\max\{|\sharp_\Box^i|, |\sharp_\Box^j|\}}{\gamma_{max}}.$$

*Proof.* The proof is based on a combinatorial argument. Let $\sharp_\Box^i = \{k_1, \ldots, k_r\}$ and let $\sharp_\Box^m = \{k_1, \ldots, k_\ell\}$, with $\ell < r$. By definition there exists $\varphi : \{k_1, \ldots, k_\ell\} \to \{w_1, \ldots, w_\ell\}$, with $k_i \sim w_i$ and $w_i \in \sharp_\Box^j$, for $1 \leq i \leq \ell$. Given $k \in \sharp_\Box^j \setminus \{k_1, \ldots, k_\ell\}$, then there are no $w \in \sharp_\Box^j \setminus \{w_1, \ldots, w_\ell\}$ such that $k \sim w$, otherwise we could extend $\varphi$ by setting $k \mapsto \varphi(k) := w$ and we would then get $|\sharp_\Box^m| = \ell + 1$. Accordingly, we have

$$\sum_{s=1}^{\ell} (A_{w_s} \cdot (A_i - A_j \odot A_i)) - 1) \geq |\sharp_\Box^i|,$$

which implies

$$\gamma_{max} |\sharp_\Box^m| \equiv \gamma_{max} \ell \geq \sum_{s=1}^{\ell} (A_{w_s} \cdot (A_i - A_j \odot A_i)) - 1) \geq |\sharp_\Box^i|.$$

$\square$

# C    EXISTING CURVATURE CANDIDATES

**Ollivier Ricci curvature**    For $i \in V$ and $\alpha \in [0, 1)$ we define a probability measure on $B_1(i)$ by:

$$\mu_i^\alpha : j \mapsto \begin{cases} \alpha, & j = i \\ \frac{1-\alpha}{d_i}, & j \in S_1(i), \\ 0, & \text{otherwise.} \end{cases}$$

Before we introduce the Ollivier curvature, we recall that the *transportation distance* between two finitely supported probability measures as above can be computed as

$$W_1(\mu_i^\alpha, \mu_j^\alpha) := \inf_M \sum_{k \in S_1(i)} \sum_{w \in S_1(j)} M_{kw} d_G(k, w),$$

where $d_G(\cdot, \cdot)$ is the geodesic distance on the graph and the infimum is taken over all matrices $M$ satisfying the marginal constraints:

$$\sum_{k \in S_1(i)} M_{kw} = \mu_j^\alpha(w), \qquad \sum_{w \in S_1(j)} M_{kw} = \mu_i^\alpha(k).$$

We are now ready to define the Ollivier Ricci curvature: the formulation below is due to Lin et al. (2011).

**Definition 5.** *Given $i \sim j$ we define the $\alpha$-Ollivier curvature by*

$$\kappa_\alpha(i, j) := 1 - W_1(\mu_i^\alpha, \mu_j^\alpha). \tag{7}$$

*Since $\kappa_\alpha (1 - \alpha)^{-1}$ is increasing and bounded the quantity below is well-defined:*

$$\kappa(i, j) := \lim_{\alpha \to 1} \frac{1 - W_1(\mu_i^\alpha, \mu_j^\alpha)}{1 - \alpha}. \tag{8}$$

**Forman Ricci curvature** In the following we report a formula for the *augmented* Forman Ricci curvature on unweighted graphs Samal et al. (2018). We also note that Forman curvature on graphs has also been studied in Sreejith et al. (2016); Weber et al. (2018).

**Definition 6.** *For any edge $i \sim j$ the augmented Forman curvature is given by*

$$F(i, j) := 4 - d_i - d_j + 3|\sharp_\Delta|.$$

We note that such formulation of curvature does not distinguish contributions coming from 4-cycles. In fact, for the orthogonal grid with degree $d \geq 4$, Forman Ricci curvature is equal to $2(2 - d) < 0$. This does not reflect that the $r$-hop neighbourhood for such a graph grows polynomially in $r$.

We conclude this appendix by reporting a lower bound for the Ollivier Ricci curvature derived in Jost & Liu (2014). We recall that $\kappa_\alpha$, with $\alpha \in [0, 1)$ was defined in equation 7.

**Theorem 12** (Jost & Liu (2014)). *For any edge $i \sim j$, with $d_i \leq d_j$, the following bound is satisfied:*

$$\kappa_0(i, j) \geq \Phi(i, j) := -\left(1 - \frac{1}{d_i} - \frac{1}{d_j} - \frac{|\sharp_\Delta|}{d_j}\right)_+ - \left(1 - \frac{1}{d_i} - \frac{1}{d_j} - \frac{|\sharp_\Delta|}{d_i}\right)_+ + \frac{|\sharp_\Delta|}{d_j}.$$

# D Proofs of results in Section 3

We first recall our definition of Balanced Forman:

**Definition 7.** *For any edge $i \sim j$ we let $\mathrm{Ric}(i, j)$ be zero if $\min\{d_i, d_j\} = 1$, otherwise*

$$\mathrm{Ric}(i, j) := \frac{2}{d_i} + \frac{2}{d_j} - 2 + 2\frac{|\sharp_\Delta|}{\max\{d_i, d_j\}} + \frac{|\sharp_\Delta|}{\min\{d_i, d_j\}} + \frac{(\gamma_{max})^{-1}}{\max\{d_i, d_j\}}(|\sharp_\Box^i| + |\sharp_\Box^j|) \quad (9)$$

*where the last term is set to be zero if $|\sharp_\Box^i|$ (and hence $|\sharp_\Box^j|$) is zero.*

We also extend the previous definition to the weighted case. In this setting we let $G = (V, E, \omega)$ be a simple, locally finite, undirected graph with normalized weights. We first report the formula for the augmented Forman in the weighted case Samal et al. (2018):

$$F(i, j) = \omega(i) + \omega(j) + \sum_{k \in S_1(i) \cap S_1(j)} \frac{\omega_{ij}^2}{\omega_\Delta}$$

$$- \sum_{k \in S_1(i) \backslash S_1(j)} \omega(i)\sqrt{\frac{\omega_{ij}}{\omega_{ik}}} - \sum_{k \in S_1(j) \backslash S_1(i)} \omega(j)\sqrt{\frac{\omega_{ij}}{\omega_{jk}}},$$

where $\omega_\Delta$ is taken to be the Heron formula for the area of a triangle while $\omega(\cdot)$ denotes some weighting scheme for the nodes as well. We propose a similar definition for the weighted case, which reduces to the one discussed above in the combinatorial setting. We recall that $W$ is the weighted adjacency matrix while $A$ is the combinatorial one. Moreover, we write $A_j^i = (A_j - A_i)_+$ and similarly for $W_j^i$.

**Definition 8.** *For any pair of adjacent nodes $i, j \in V$ we define $\mathrm{Ric}(i, j)$ to be 0 if $\min\{|d_i|, |d_j|\} = 1$, otherwise we set*

$$\mathrm{Ric}(i, j) := \frac{1}{d_i}\left(1 - \sum_{k \in Q_i}\sqrt{\frac{\omega_{ij}}{\omega_{ik}}}\right) + \frac{1}{d_j}\left(1 - \sum_{k \in Q_j}\sqrt{\frac{\omega_{ij}}{\omega_{jk}}}\right)$$

$$+ \frac{1}{\max\{d_i, d_j\}}\sum_{k \in S_1(i) \cap S_1(j)} \frac{\omega_{ij}^2}{\omega_\Delta}$$

$$+ \sum_{k \in \sharp_\Box^i} \frac{\omega_{ij}}{\omega_\Box}\left(\frac{\omega_{ij}(\gamma_{max})^{-1}}{\max\{d_i, d_j\}} - \frac{\sqrt{\omega_{ij}\nu_\Box}}{d_i}\right)$$

$$+ \sum_{k \in \sharp_\Box^j} \frac{\omega_{ij}}{\omega_\Box}\left(\frac{\omega_{ij}(\gamma_{max})^{-1}}{\max\{d_i, d_j\}} - \frac{\sqrt{\omega_{ij}\nu_\Box}}{d_j}\right),$$

*with*

$$\omega_\Delta := \frac{1}{3}(\omega_{ij}^2 + \omega_{ik}^2 + \omega_{jk}^2), \quad z \in S_1(i) \cap S_1(j),$$

*and, for a given* $k \in \sharp_\Box^i$,

$$\nu_\Box := \frac{W_z \cdot A_j^i}{A_k \cdot A_j^i}, \quad \omega_\Box = \frac{1}{4}\left(\omega_{ij}^2 + \omega_{ik}^2 + \nu_\Box^2 + \frac{A_k \cdot W_j^i}{A_k \cdot A_j^i}\right).$$

**Important convention.** Without losing generality, in the following we always assume that $1 \leq d_i \leq d_j$. In particular, we write $d_i \doteq d$ and $d_j = d + s$, for some $s \geq 0$, omitting to specify that both $d$ and $s$ are of course depending on $i$ and $j$. Moreover, from now on we only focus on the unweighted case.

We can now prove our main comparison theorem.

**Theorem 2.** *Given an unweighted graph $G$, for any edge $i \sim j$ we have $\kappa(i, j) \geq \mathrm{Ric}(i, j)$.*

*Proof.* We stick to the aforementioned convention: $d_i := d \leq d_j = d + s$, $s \geq 0$. The strategy of the proof amounts to finding a transportation plan providing an upper bound for $W_1(\mu_i^\alpha, \mu_j^\alpha)$ and hence a lower bound for the curvature $\kappa(i, j)$. In particular, we consider plans moving the mass $\mu_i^\alpha$ from $B_1(i)$ to $B_1(j)$.

If $d = 1$, then the optimal transport plan consists of moving the mass $\alpha$ from $i$ to $j$ and the remaining mass $1 - \alpha$ on $j$ to $S_1(j)$. This yields a unit Wasserstein distance between $\mu_i^\alpha$ and $\mu_j^\alpha$ and hence zero Ollivier curvature $\kappa(i, j)$, which coincides with the value of balanced Forman $\mathrm{Ric}(i, j)$.

Assume now that $d \geq 2$. A (possibly non-optimal) transport plan from $\mu_i^\alpha$ to $\mu_j^\alpha$ is given by:

(i) Move mass $(1 - \alpha)/(d + s)$ from each node $k \in \sharp_\Box^m \subset \sharp_\Box^i$ to its unique image in $\sharp_\Box^j$ under a bijection $\varphi$ as per definition of $\sharp_\Box^m$.

(ii) The remaining mass on each node $k \in \sharp_\Box^m$ will need to travel by at most distance 3 to $S_1(j)$.

(iii) The extra-mass $(1 - \alpha)(1/d - 1/(d + s))$ on each common neighbour $k \in \sharp_\Delta$ will need to travel by at most distance 2 to $S_1(j)$.

(iv) Move the mass $(1 - \alpha)/d$ from $j$ to $S_1(j)$.

(v) Move the mass $\alpha$ from $i$ to $j$. This leaves left-over mass $(1 - \alpha)/(d + s)$ at $i$ from the distribution $\mu_j^\alpha$. This mass can be compensated from mass in $S_1(i)$ which is at distance one.

(vi) Finally, we move the mass $(1 - \alpha)/d$ of any untouched node in $S_1(i)$ to $S_1(j)$ along a path of length lesser or equal than three. Note that the remaining mass is equal to $((1 - \alpha)/d)(d - 1 - |\sharp_\Delta| - |\sharp_\Box^m|) - (1 - \alpha)(d + s)$, where the last terms comes from (v).

If we sum all the contributions we find

$$W_1(\mu_i^\alpha, \mu_j^\alpha) \leq (1 - \alpha)\left(\frac{|\sharp_\Box^m|}{d + s} + 3|\sharp_\Box^m|\left(\frac{1}{d} - \frac{1}{d + s}\right) + 2|\sharp_\Delta|\left(\frac{1}{d} - \frac{1}{d + s}\right) + \frac{1}{d}\right) + \alpha + \frac{1 - \alpha}{d + s}$$

$$+ 3(1 - \alpha)\left(\frac{1}{d}(d - 1 - |\sharp_\Delta| - |\sharp_\Box^m|) - \frac{1}{d + s}\right)$$

$$= (1 - \alpha)\left(\frac{1}{d + s}(-2|\sharp_\Delta| - 2|\sharp_\Box^m| - 2) + \frac{1}{d}(-|\sharp_\Delta| - 2) + 2\right) + 1$$

$$= (1 - \alpha)\left(\frac{1}{d + s}\left(-3|\sharp_\Delta| - \frac{s}{d}|\sharp_\Delta| - 2|\sharp_\Box^m| - 4 - 2\frac{s}{d} + 2(d + s)\right)\right) + 1.$$

Therefore we have

$$\kappa(i, j) = \lim_{\alpha \to 1} \frac{1 - W(\mu_i^\alpha, \mu_j^\alpha)}{1 - \alpha} \geq \left(\frac{1}{d + s}\left(3|\sharp_\Delta| + \frac{s}{d}|\sharp_\Delta| + 2|\sharp_\Box^m| + 4 + 2\frac{s}{d} - 2(d + s)\right)\right).$$

By using Lemma 11, we can bound the right hand side as

$$\kappa(i,j) \geq \left( \frac{1}{d+s} \left( 3|\sharp_\Delta| + \frac{s}{d}|\sharp_\Delta| + (\gamma_{\max})^{-1}(|\sharp_\square^i| + |\sharp_\square^j|) + 4 + 2\frac{s}{d} - 2(d+s) \right) \right) = \mathrm{Ric}(i,j)$$

which completes the proof. $\qquad\square$

**Remark 13.** *By inspection* $\mathrm{Ric}(i,j) \geq \Phi(i,j)$, *with* $\Phi(i,j)$ *as in Theorem 12. We have three cases:*

(i) $\Phi(i,j) = 2/d + 2/(d+s) - 2 + 2|\sharp_\Delta|/(d+s) + |\sharp_\Delta|/d \leq \mathrm{Ric}(i,j)$, *because* $\mathrm{Ric}(i,j)$ *takes into account the positive contribution of 4-cycles as well.*

(ii) $\Phi(i,j) = -1 + 1/d + 1/(d+s) + 2|\sharp_\Delta|/(d+s)$, *which happens iff*

$$d + s - 2 - \frac{s}{d} - |\sharp_\Delta| > 0$$

*and*

$$d + s - 2 - \frac{s}{d} - |\sharp_\Delta| - \frac{s}{d}|\sharp_\Delta| \leq 0.$$

*From the previous inequalities we derive*

$$\mathrm{Ric}(i,j) \geq \Phi(i,j) + \frac{1}{d+s} \left( 2 - d - s + |\sharp_\Delta| + \frac{s}{d}|\sharp_\Delta| + \frac{s}{d} \right) \geq \Phi(i,j).$$

(iii) $\Phi(i,j) = |\sharp_\Delta|/(d+s)$ *which is equivalent to*

$$d + s - 2 - \frac{s}{d} - |\sharp_\Delta| \leq 0.$$

*In this case we have*

$$\mathrm{Ric}(i,j) \geq \Phi(i,j) + \frac{s}{d}\frac{|\sharp_\Delta|}{d+s}$$

**Corollary 3.** *If* $\mathrm{Ric}(i,j) \geq k > 0$ *for any edge* $i \sim j$, *then there exists a polynomial* $P$ *such that*

$$|B_r(i)| \leq P(r), \quad \forall i \in V.$$

*Proof.* This follows immediately from Theorem 2 and a Bishop-Gromov type of result for discrete Ollivier curvature on graphs Paeng (2012). $\qquad\square$

To address the proof of Theorem 4, we first need the Lemma below.

**Lemma 14.** *Given* $i \sim j$, *with* $d_i \leq d_j$, *if* $\mathrm{Ric}(i,j) \leq -2 + \delta$, *for some* $0 < \delta < (1 + \gamma_{max})^{-1}$, *then*

$$\frac{|Q_j|}{|\sharp_\Delta| + 1} > \delta^{-1}.$$

*Proof.* According to our convention we let $d_i = d$ and $d_j = d + s$, for $s \geq 0$. We also recall that $Q_j = S_1(j) \setminus (\sharp_\Delta \cup \sharp_\square^j \cup \{i\})$. If we multiply equation 3 by $d_j = d+s$, we see that $\mathrm{Ric}(i,j) \leq -2+\delta$ iff

$$4 + 2\frac{s}{d} + 3|\sharp_\Delta| + \frac{s}{d}|\sharp_\Delta| + \gamma_{\max}^{-1}(|\sharp_\square^i| + |\sharp_\square^j|) \leq \delta(1 + |\sharp_\Delta| + |\sharp_\square^j| + |Q_j|).$$

By assumption $|\sharp_\square^j|(\gamma_{\max}^{-1} - \delta) \geq 0$, meaning that

$$\delta(1 + |\sharp_\Delta| + |Q_j|) \geq 4 + 2\frac{s}{d} + 3|\sharp_\Delta| + \frac{s}{d}|\sharp_\Delta| \geq 4 + 3|\sharp_\Delta|.$$

Therefore, we conclude

$$\frac{|Q_j|}{|\sharp_\Delta| + 1} \geq \frac{3}{\delta} - 1.$$

$\qquad\square$

**Theorem 4.** *Consider a MPNN as in equation 1. Let* $i \sim j$ *with* $d_i \leq d_j$ *and assume that:*

(i) $|\nabla\phi_\ell| \le \alpha$ and $|\nabla\psi_\ell| \le \beta$ for each $0 \le \ell \le L-1$, with $L \ge 2$ the depth of the MPNN.

(ii) There exists $\delta$ s.t. $0 < \delta < (\max\{d_i, d_j\})^{-\frac{1}{2}}$, $\delta < \gamma_{max}^{-1}$, and $\mathrm{Ric}(i,j) \le -2 + \delta$.

Then there exists $Q_j \subset S_2(i)$ satisfying $|Q_j| > \delta^{-1}$ and for $0 \le \ell_0 \le L-2$ we have

$$\frac{1}{|Q_j|} \sum_{k \in Q_j} \left| \frac{\partial h_k^{(\ell_0+2)}}{\partial h_i^{(\ell_0)}} \right| < (\alpha\beta)^2 \delta^{\frac{1}{4}}. \tag{4}$$

*Proof.* As usual we let $d_i := d$ and $d_j := d + s$, for some $s \ge 0$. We first observe that from the requirement $\delta^2(d+s) \le 1$ in (ii), we derive $\mathrm{Ric}(i,j) \le -2 + \delta$ iff

$$4 + 2\frac{s}{d} + 3|\sharp_\Delta| + \frac{s}{d}|\sharp_\Delta| + \gamma_{max}^{-1}(|\sharp_\Box^i| + |\sharp_\Box^j|) \le \delta(d+s).$$

Therefore we have

$$\delta|\sharp_\Delta| \left(3 + \frac{s}{d}\right) \le \delta^2(d+s),$$

meaning that

$$\delta|\sharp_\Delta| \le 1. \tag{10}$$

From now on we let $Q_j$ denote again the complement $S_1(j) \setminus (S_1(i) \cup \sharp_\Box^j \cup \{i\})$. Without loss of generality we set $\ell_0 = 0$ and hence $h_i^{(0)} = x_i$; the very same proof applies to any other choice of $\ell_0$. Given $k \in Q_j$, since $k \in S_2(i)$, we can apply Lemma 1 and derive

$$\left| \frac{\partial h_k^{(2)}}{\partial x_i} \right| \le (\alpha\beta)^2 (\hat{A})_{ik}^2. \tag{11}$$

We may expand the power of the augmented normalized adjacency matrix as

$$(\hat{A})_{ik}^2 = \frac{1}{\sqrt{(d_k+1)(d_i+1)}} \sum_{w \in S_1(k) \cap S_1(i)} \frac{1}{d_w + 1}.$$

If we introduce the set $\hat{Q}_j = \{k \in Q_j : \sigma_{ik} > 1\}$, we can then write

$$\sum_{k \in Q_j} (\hat{A})_{ik}^2 = \sum_{k \in Q_j} \frac{1}{\sqrt{(d_k+1)(d_i+1)}} \sum_{w \in S_1(k) \cap S_1(i)} \frac{1}{d_w + 1} =$$

$$= \frac{1}{\sqrt{d_i+1}} \left( \sum_{k \in Q_j} \frac{1}{\sqrt{d_k+1}} \frac{1}{d_j+1} + \sum_{k \in \hat{Q}_j} \frac{1}{\sqrt{d_k+1}} \sum_{w \in S_1(k) \cap S_1(i) \cap S_1(j)} \frac{1}{d_w+1} \right) \tag{12}$$

where in the last equality we have again used the fact that $k \in \hat{Q}_j$ iff there is $w \in S_1(k) \cap S_1(i) \cap S_1(j)$. To avoid heavy notations, we introduce $V_k := S_1(k) \cap S_1(i) \cap S_1(j)$. Let us first focus on the first sum in equation 12. We have

$$\frac{1}{\sqrt{d_i+1}} \sum_{k \in Q_j} \frac{1}{\sqrt{d_k+1}} \frac{1}{d_j+1} \le \frac{1}{\sqrt{d_i+1}} |Q_j| \frac{1}{d_j+1} \le \frac{1}{\sqrt{d_i+1}} \le 1. \tag{13}$$

We now consider the second sum in equation 12. We assume $|\sharp_\Delta| \ge 1$, otherwise $\hat{Q}_j = \emptyset$. We let

$$\Omega := \left\{ w \in \sharp_\Delta : d_w < \frac{1}{C} \frac{|\hat{Q}_j|}{|\sharp_\Delta|} + \frac{2}{C} \right\}$$

for some $C > 0$ to be chosen below. Then, the second sum in equation 12 can be split as

$$\sum_{k \in \hat{Q}_j} \frac{1}{\sqrt{d_k+1}} \left( \sum_{w \in V_k \cap \Omega} \frac{1}{d_w+1} + \sum_{w \in V_k \setminus \Omega} \frac{1}{d_w+1} \right). \tag{14}$$

Since any $w \in V_k$ has degree at least three, we can bound the first term in equation 14 as

$$\sum_{k \in \hat{Q}_j} \frac{1}{\sqrt{d_k + 1}} \left( \sum_{w \in V_k \cap \Omega} \frac{1}{d_w + 1} \right) \leq \sum_{k \in \hat{Q}_j} \frac{1}{\sqrt{d_k + 1}} \frac{|V_k \cap \Omega|}{4}.$$

We now observe that

$$\sum_{k \in \hat{Q}_j} \frac{|V_k \cap \Omega|}{\sqrt{d_k + 1}} \leq \sum_{k \in \hat{Q}_j} |V_k \cap \Omega| = \left| \left\{ (k, w) \in E : \ k \in \hat{Q}_j, w \in V_k \cap \Omega \right\} \right| \leq \left( \max_{w \in \Omega} d_w \right) |\Omega|.$$

Since $d_w \leq (1/C)|\hat{Q}_j|/|\sharp_\Delta| + 2/C$ for any $w \in \Omega$ we see that the first term in equation 14 can be bounded by

$$\sum_{k \in \hat{Q}_j} \frac{1}{\sqrt{d_k + 1}} \sum_{w \in V_k \cap \Omega} \frac{1}{d_w + 1} \leq \sum_{k \in \hat{Q}_j} \frac{1}{\sqrt{d_k + 1}} \frac{|V_k \cap \Omega|}{4}$$

$$\leq \left( \frac{1}{C} \frac{|\hat{Q}_j|}{|\sharp_\Delta|} + \frac{2}{C} \right) \frac{|\Omega|}{4} \leq \left( \frac{1}{C} \frac{|\hat{Q}_j|}{|\sharp_\Delta|} + \frac{2}{C} \right) \frac{|\sharp_\Delta|}{4}. \tag{15}$$

by definition of $\Omega$. We now bound the second term in equation 14 as

$$\sum_{k \in \hat{Q}_j} \frac{1}{\sqrt{d_k + 1}} \sum_{w \in V_k \setminus \Omega} \frac{1}{d_w + 1} \leq \sum_{k \in \hat{Q}_j} \frac{1}{\sqrt{d_k + 1}} \frac{C|\sharp_\Delta|}{|\hat{Q}_j|} |V_k \setminus \Omega|$$

where we have used that $d_w^{-1} \leq C|\sharp_\Delta|/|\hat{Q}_j|$ if $w \in V_k \setminus \Omega$. Since

$$\frac{|V_k \setminus \Omega|}{\sqrt{d_k + 1}} \leq \frac{|V_k|}{\sqrt{|S_1(k)|}} \leq \frac{|V_k|}{\sqrt{|V_k|}} \leq \sqrt{|V_k|} \leq \sqrt{|\sharp_\Delta|}$$

we see that

$$\sum_{k \in \hat{Q}_j} \frac{1}{\sqrt{d_k + 1}} \frac{C|\sharp_\Delta|}{|\hat{Q}_j|} |V_k \setminus \Omega| \leq \frac{C|\sharp_\Delta|}{|\hat{Q}_j|} \sqrt{|\sharp_\Delta|}|\hat{Q}_j| = C|\sharp_\Delta|^{\frac{3}{2}}. \tag{16}$$

We are now ready to complete the proof of the theorem. According to equation 11 it suffices to show that

$$\frac{1}{|Q_j|} \sum_{k \in Q_j} (\hat{A}^2)_{ik} \leq \delta^{1/4}.$$

From equation 12 and equation 13 we derive that the left hand side of the equation above is bounded by

$$\frac{1}{|Q_j|} \sum_{k \in Q_j} (\hat{A}^2)_{ik} \leq \frac{1}{|Q_j|} + \frac{1}{|Q_j|} \left( \sum_{k \in \hat{Q}_j} \frac{1}{\sqrt{d_k + 1}} \sum_{w \in V_k} \frac{1}{d_w + 1} \right)$$

$$\leq \delta + \frac{1}{|Q_j|} \left( \sum_{k \in \hat{Q}_j} \frac{1}{\sqrt{d_k + 1}} \sum_{w \in V_k} \frac{1}{d_w + 1} \right)$$

where in the last inequality we have used Lemma 14 to bound $|Q_j|^{-1}$ by $\delta$. In particular we note that if $\sharp_\Delta = \emptyset$ then $\hat{Q}_j = \emptyset$, and the bound would be simply controlled by $\delta$ as claimed. When $|\sharp_\Delta| > 0$, we can use equation 15 and equation 16 to estimate the second term from above by

$$\frac{1}{|Q_j|} \left( \sum_{k \in \hat{Q}_j} \frac{1}{\sqrt{d_k + 1}} \sum_{w \in V_k} \frac{1}{d_w + 1} \right) \leq$$

$$\frac{1}{|Q_j|} \left( \left( \frac{1}{C} \frac{|\hat{Q}_j|}{|\sharp_\Delta|} + \frac{2}{C} \right) \frac{|\sharp_\Delta|}{4} \right) + \frac{1}{|Q_j|} \left( C|\sharp_\Delta|^{\frac{3}{2}} \right) \leq \frac{1}{4} \left( \frac{1}{C} + \frac{2|\sharp_\Delta|}{C|Q_j|} \right) + \frac{C|\sharp_\Delta|}{|Q_j|} \sqrt{|\sharp_\Delta|}.$$

By applying Lemma 14 we get

$$\frac{1}{4}\left(\frac{1}{C} + \frac{2|\sharp_\Delta|}{C|Q_j|}\right) + \frac{C|\sharp_\Delta|}{|Q_j|}\sqrt{|\sharp_\Delta|} \leq \frac{1}{4}\left(\frac{1}{C} + 2\frac{\delta}{C}\right) + C\delta\sqrt{|\sharp_\Delta|}.$$

We now choose $C = \delta^{-1/4}$, so that the previous quantity can be bounded by

$$\frac{1}{4}\left(\frac{1}{C} + 2\frac{\delta}{C}\right) + C\delta\sqrt{|\sharp_\Delta|} \leq \frac{1}{4}\left(\delta^{\frac{1}{4}} + 2\delta^{\frac{5}{4}}\right) + \delta^{\frac{1}{4}}\sqrt{\delta|\sharp_\Delta|} \leq \frac{1}{4}\left(\delta^{\frac{1}{4}} + 2\delta^{\frac{5}{4}}\right) + \delta^{\frac{1}{4}}$$

where in the last inequality we have used equation 10. Therefore, we have shown that

$$\frac{1}{|Q_j|}\sum_{k \in Q_j}(\hat{A}^2)_{ik} \leq \delta + \frac{1}{4}\left(\delta^{\frac{1}{4}} + 2\delta^{\frac{5}{4}}\right) + \delta^{\frac{1}{4}} \leq 3\delta^{\frac{1}{4}}$$

where we have used that $\delta < 1$. This completes the proof (once we absorb the extra factor 3 in the constant $\alpha\beta$ in equation 11). $\square$

**Remark 15.** *The requirement $\delta\sqrt{\max\{d_i, d_j\}} < 1$ can be replaced by a more general bound $\delta\sqrt{\max\{d_i, d_j\}} < r$. The argument above extends to this case up to renaming the constant $\alpha\beta$ in the statement so to include an extra factor $r$.*
*We note that the condition $\delta\max\{d_i, d_j\} < r$ would be stronger than the one appearing in (ii) of Theorem 4. In this regard, we recall that for a $d$-tree the curvature satisfies $\mathrm{Ric}(i, j) = -2 + \frac{4}{d}$.*

We can also prove Proposition 5:

**Proposition 5.** *If $\mathrm{Ric}(i, j) \geq k > 0$ for all $i \sim j$, then $\lambda_1/2 \geq h_G \geq \frac{k}{2}$.*

*Proof.* This follows as an immediate Corollary of Theorem 2 and (Lin et al., 2011, Theorem 4.2). $\square$

**Betweenness centrality to measure bottleneck.** In equation 2 we have derived how the topology of the graph affects the dependence of the hidden node representation $h_i^{(r+1)}$ on the input feature $x_s$, for nodes $i, s$ at distance $r + 1$. We note that in this case $\hat{A}_{is}^{r+1}$ is exactly measuring the number of minimal paths from $i$ to $s$. If the receptive field $B_{r+1}(i)$ is a binary tree, then we have seen that the entry of the power matrix decays exponentially. The reason for such decay stems from the existence of exponentially many nodes in the receptive field combined with the lack of multiple minimal paths (shortcuts). When such conditions hold, most of the minimal paths go through the same nodes, which is exactly what happens for the tree where each node is in the minimal paths between different branches. Since the frequency in which a node appears in the minimal path of distinct pairs of nodes is measured by the *betweenness centrality* Freeman (1977), we propose a topological characterization of the 'bottleneckedness' of a graph as follows:

**Definition 9 (bottleneck).** *The bottleneck-value of $G$ is $b_G := \frac{1}{n}\sum_{i=1}^n b(i)$, where $b(i)$ denotes the betweenness centrality on node $i$.*

From a standard combinatorial argument it follows that if $G$ is connected, then

$$b_G = \frac{1}{n}\sum_{i,j}(d_G(i, j) - 1). \tag{17}$$

We note that $b_G = 0$ iff $G$ is the complete graph $K_n$. Therefore, $b_G$ determines how far the given topology is from $K_n$, with the latter representing the limit case of a fully connected layer Alon & Yahav (2021) where no bottleneck may occur as any pair of nodes would be neighbours. This further supports our intuition that the betweenness centrality is a good topological candidate for providing a global measurement of bottleneckedness in the graph.

It also follows from equation 17 that any update to the graph topology consisting of edge additions would decrease $b_G$ and thus reduce the bottleneck. The quantity $b_G$ is *global* in nature though and hence lacks robustness. As a pedagogical example, consider a barbell $G(m, 2r + 1)$, with $m$ the size of the two cliques joined by a path of length $2r + 1$ and focus on the edge $i \sim j$ in the middle of such path. Nodes $i$ and $j$ are central to the graph, in the sense that most minimal paths go through them and indeed their betweenness centrality is $b(i) = b(j) = (m + r)^2 + (m + r)$. If now we add a

*single* edge joining the two cliques $K_m$, the values $b(i)$ and $b(j)$ decrease dramatically by $\Omega(m^2 + r)$. Since the operation is non-local, we see that the representation $h_i^{(\ell)}$ of any MPNN is *unaffected* by the edge addition for any $\ell \in (0, r)$, and similarly for $j$. Eventually, if we keep adding edges, the receptive fields $B_s(i)$ will be affected for small values of $s$ as well: the drawback of such approach is that the resulting adjacency may be significantly different. Conversely, the curvature provides a more precise, local and hence robust way of controlling the bottleneck and hence the over-squashing problem. Nonetheless, we relate the betweenness centrality to the Jacobian of the hidden features.

**Theorem 16.** *Given $i \sim j$, let $\Omega_j := S_1(i) \cap S_1(j) \cup \{j\}$. If $\mathrm{Ric}(i, j) \leq -2 + \delta$, for $0 < \delta < (1 + \gamma_{max})^{-1}$, then*

$$\frac{1}{|\Omega_j|} \sum_{k \in \Omega_j} b(k) \geq \delta^{-1}.$$

*Proof.* We rewrite the quantity in the statement as

$$\frac{1}{|\sharp_\triangle| + 1} \left( \sum_{k \in \sharp_\triangle} b(k) + b(j) \right).$$

By definition, given a node $k \in V$, the betweenness centrality of $k$ is given by

$$b(k) := \sum_{s,t \in V : s \neq k, t \neq k} \frac{\sigma_{st}(k)}{\sigma_{st}}$$

where $\sigma_{st}$ is the number of minimal paths between $s$ and $t$ while $\sigma_{st}(k)$ is the number of minimal paths from $s$ to $t$ passing through $k$. For convenience, we introduce the set $\hat{Q}_j \subset Q_j$ defined by

$$\hat{Q}_j := \{w \in Q_j : \sigma_{iw} > 1\}.$$

Equivalently, $\hat{Q}_j$ consists of those nodes in $S_1(j) \setminus S_1(i)$ which form a 4-cycle based at $i \sim j$ with a diagonal inside. Indeed, if $w \in Q_j$ and $\sigma_{iw} > 1$, then there exists more than one minimal path between $i$ and $w$, in addition to the one passing through node $j$. For any such path there exists $k \in S_1(i) \cap S_1(w)$. Since $w \in Q_j$ and $Q_j \cap \sharp_\square^j = \emptyset$, we derive that $k \in S_1(j)$ as well. We then get

$$\sum_{k \in \sharp_\triangle} b(k) = \sum_{k \in \sharp_\triangle} \sum_{s,t \in V : s \neq k, t \neq k} \frac{\sigma_{st}(k)}{\sigma_{st}} \geq \sum_{k \in \sharp_\triangle} \sum_{w \in \hat{Q}_j} \frac{\sigma_{iw}(k)}{\sigma_{iw}} = \sum_{w \in \hat{Q}_j} \frac{1}{\sigma_{iw}} \sum_{k \in \sharp_\triangle} \sigma_{iw}(k).$$

By summing $\sigma_{iw}(k)$ for all $k \in \sharp_\triangle$ we obtain all the 2-long minimal paths between $i$ and $w$ with the exception of the one passing through $j$:

$$\sum_{k \in \sharp_\triangle} b(k) \geq \sum_{w \in \hat{Q}_j} \frac{1}{\sigma_{iw}}(\sigma_{iw} - 1). \tag{18}$$

On the other hand we also have

$$b(j) = \sum_{s,t \in V : s \neq j, t \neq j} \frac{\sigma_{st}(j)}{\sigma_{st}} \geq \sum_{z \in Q_j} \frac{\sigma_{iz}(j)}{\sigma_{iz}} = \sum_{z \in Q_j} \frac{1}{\sigma_{iz}}. \tag{19}$$

By combining equation 18 and equation 19 we finally get

$$\frac{1}{|\sharp_\triangle| + 1} \left( \sum_{k \in \sharp_\triangle} b(k) + b(j) \right) \geq \frac{1}{|\sharp_\triangle| + 1} \left( \sum_{w \in \hat{Q}_j} \frac{1}{\sigma_{iw}}(\sigma_{iw} - 1) + \sum_{z \in Q_j} \frac{1}{\sigma_{iz}} \right)$$

$$= \frac{1}{|\sharp_\triangle| + 1} \left( |\hat{Q}_j| + \sum_{z \in Q_j \setminus \hat{Q}_j} \frac{1}{\sigma_{iz}} \right)$$

$$= \frac{|Q_j|}{|\sharp_\triangle| + 1},$$

where in the last equality we have used that by definition $\sigma_{iz} = 1$ for all $z \in Q_j \setminus \hat{Q}_j$. By Lemma 14 the last quantity is larger than $\delta^{-1}$. $\qquad \square$

# E  PROOFS OF RESULTS IN SECTION 4

**Theorem 6.** *Let $S \subset V$ with $vol(S) \leq vol(G)/2$. Then $h_{S,\alpha} \leq \left(\frac{1-\alpha}{\alpha}\right) \frac{d_{\mathrm{avg}}(S)}{d_{\min}(S)} h_S$, where $d_{\mathrm{avg}}(S)$ and $d_{\min}(S)$ are the average and minimum degree on $S$, respectively.*

*Proof.* Given a signal $f : V \to \mathbb{R}$ on the vertex set and $U \subset V$, analogously to Chung (2007), we introduce the notation

$$f(U) := \sum_{i \in U} f(i).$$

Let us rewrite the new Cheeger constant $h_{S,\alpha}$ as follows:

$$h_{S,\alpha} = \frac{1}{|S|} \sum_{i \in S, j \in \bar{S}} (R_\alpha)_{ij} = \frac{1}{|S|} \chi_S R_\alpha(\bar{S})$$

with $\chi_S$ the characteristic function of the subset $S$, i.e. $\chi_S(i) = 1$ iff $i \in S$. Since the graph $G$ is connected, we can bound $h_{S,\alpha}$ from above as

$$\frac{1}{|S|} \chi_S R_\alpha(\bar{S}) = \frac{1}{|S|} \sum_{i \in S, j \in \bar{S}} (R_\alpha)_{ij} \leq \frac{1}{|S|} \sum_{i \in S, j \in \bar{S}} (R_\alpha)_{ij} \frac{d_i}{d_{\min}(S)} = \frac{1}{|S|} \chi_S D R_\alpha(\bar{S}) \frac{1}{d_{\min}(S)}.$$

It was proven in (Chung, 2007, Lemma 5) that

$$\chi_S D R_\alpha(\bar{S}) \leq \frac{1-\alpha}{\alpha} |\partial S|. \tag{20}$$

By applying equation equation 20 to the bound for the Cheeger constant $h_{S,\alpha}$ we finally see that

$$h_{S,\alpha} = \frac{1}{|S|} \chi_S R_\alpha(\bar{S}) \leq \frac{1}{|S|} \chi_S D R_\alpha(\bar{S}) \frac{1}{d_{\min}(S)} \leq \frac{1}{|S|} \frac{1-\alpha}{\alpha} |\partial S| \frac{1}{d_{\min}(S)}$$
$$= \frac{1}{|S|} \frac{1-\alpha}{\alpha} h_S vol(S) \frac{1}{d_{\min}(S)} = \frac{1-\alpha}{\alpha} h_S \frac{d_{\mathrm{avg}}(S)}{d_{\min}(S)}.$$

$\square$

We also report an equivalent result, again relying on (Chung, 2007, Lemma 5).

**Proposition 17.** *Let $S \subset V$ with $vol(S) \leq vol(G)/2$. For any $k \in \mathbb{N}$, there exists $S_{k,\alpha} \subset S$ with $vol(S_{k,\alpha}) \geq vol(S)(1 - (2k)^{-1})$ such that*

$$\sum_{j \in \bar{S}} (R_\alpha)_{ij} \leq k \frac{1-\alpha}{\alpha} h_S,$$

*for all $i \in S_{k,\alpha}$.*

*Proof.* Let $k \in \mathbb{N}$. By modifying slightly the argument in (Chung, 2007, Lemma 5), we derive that

$$S'_{k,\alpha} := \{i \in S : \chi_i R_\alpha(\bar{S}) \geq k \frac{1-\alpha}{\alpha} h_S\}$$

satisfies

$$\frac{1-\alpha}{2\alpha} |\partial S| \geq vol(S'_{k,\alpha}) k \frac{1-\alpha}{\alpha} h_S.$$

Therefore, we obtain

$$vol(S'_{k,\alpha}) \leq \frac{1}{k} \frac{vol(S)}{2}.$$

We then conclude that the complement of $S'_{k,\alpha}$ has volume greater or equal than $vol(S)(1 - (2k)^{-1})$, which completes the proof. $\square$

**Remark 18.** *The previous proposition shows that after sparsifying the personalized page rank operator $R_\alpha$ as suggested in Klicpera et al. (2019) by setting entries below some threshold equal to zero, there will still be only few edges connecting different communities, once again highlighting that random-walk based methods are generally not suited to tackle the graph bottleneck.*

# F  EXPERIMENTS

Our experiments in this paper are semi-supervised node classification (semi-supervised in that the graph structure provides some unlabelled information) on nine common graph learning datasets. Cornell, Texas and Wisconsin are small heterophilic datasets based on webpage networks from the WebKB dataset. Chameleon and Squirrel (Rozemberczki et al., 2021) are medium heterophilic datasets based on Wikipedia networks, along with Actor, the actor-only induced subgraph of the film-director-actor-writer network (Tang et al., 2009). Cora (McCallum et al., 2000), Citeseer (Sen et al., 2008) and Pubmed (Namata et al., 2012) are medium homophilic datasets based on citation networks. As in Klicpera et al. (2019), for all experiments we consider the largest connected component of the graph.

When splitting the data into train/validation/test sets, we first separate the data into a development set and the test set. This is done once to ensure the test set is not used for any training or hyperparameter fitting before the final evaluation. For each of the 100 random splits the development set is divided randomly into a train set and a validation set, where we train models on the train set and evaluate on the validation set. We fit hyperparameters by random search, maximising the mean accuracy across the validation sets. The accuracy then reported in Table 2 is the mean accuracy on the test set from models trained on the train sets with the chosen hyperparameters, along with a 95% confidence interval calculated by bootstrapping the test set accuracies with 1000 samples. For Cora, Citeseer and Pubmed the development set contains 1500 nodes with the rest kept for the test set, and for each random split the train set is chosen to contain 20 nodes of each class while the rest form the validation set. As this is the same method as Klicpera et al. (2019) and we use the same random seeds, we are using the same test set and expect to have comparable results. For the remaining datasets we perform a 60/20/20 split, with 20% of nodes set aside as the test set and then for each random split the remaining 80% is split into 60% training, 20% validation.

The homophily index $\mathcal{H}(G)$ proposed by Pei et al. (2019) is defined as

$$\mathcal{H}(G) = \frac{1}{|V|} \sum_{v \in V} \frac{\text{Number of } v\text{'s neighbors who have the same label as } v}{\text{Number of } v\text{'s neighbors}}. \tag{21}$$

## F.1  DATASETS

For datasets with disconnected graphs, the statistics shown here are for the largest connected component.

|  | Cornell | Texas | Wisconsin | Chameleon | Squirrel | Actor | Cora | Citeseer | Pubmed |
|---|---|---|---|---|---|---|---|---|---|
| $\mathcal{H}(G)$ | 0.11 | 0.06 | 0.16 | 0.25 | 0.22 | 0.24 | 0.83 | 0.72 | 0.79 |
| Nodes | 140 | 135 | 184 | 832 | 2186 | 4388 | 2485 | 2120 | 19717 |
| Edges | 219 | 251 | 362 | 12355 | 65224 | 21907 | 5069 | 3679 | 44324 |
| Features | 1703 | 1703 | 1703 | 2323 | 2089 | 931 | 1433 | 3703 | 500 |
| Classes | 5 | 5 | 5 | 5 | 5 | 5 | 7 | 6 | 3 |
| Directed? | Yes | Yes | Yes | Yes | Yes | Yes | No | No | No |

## F.2 DEGREE DISTRIBUTIONS

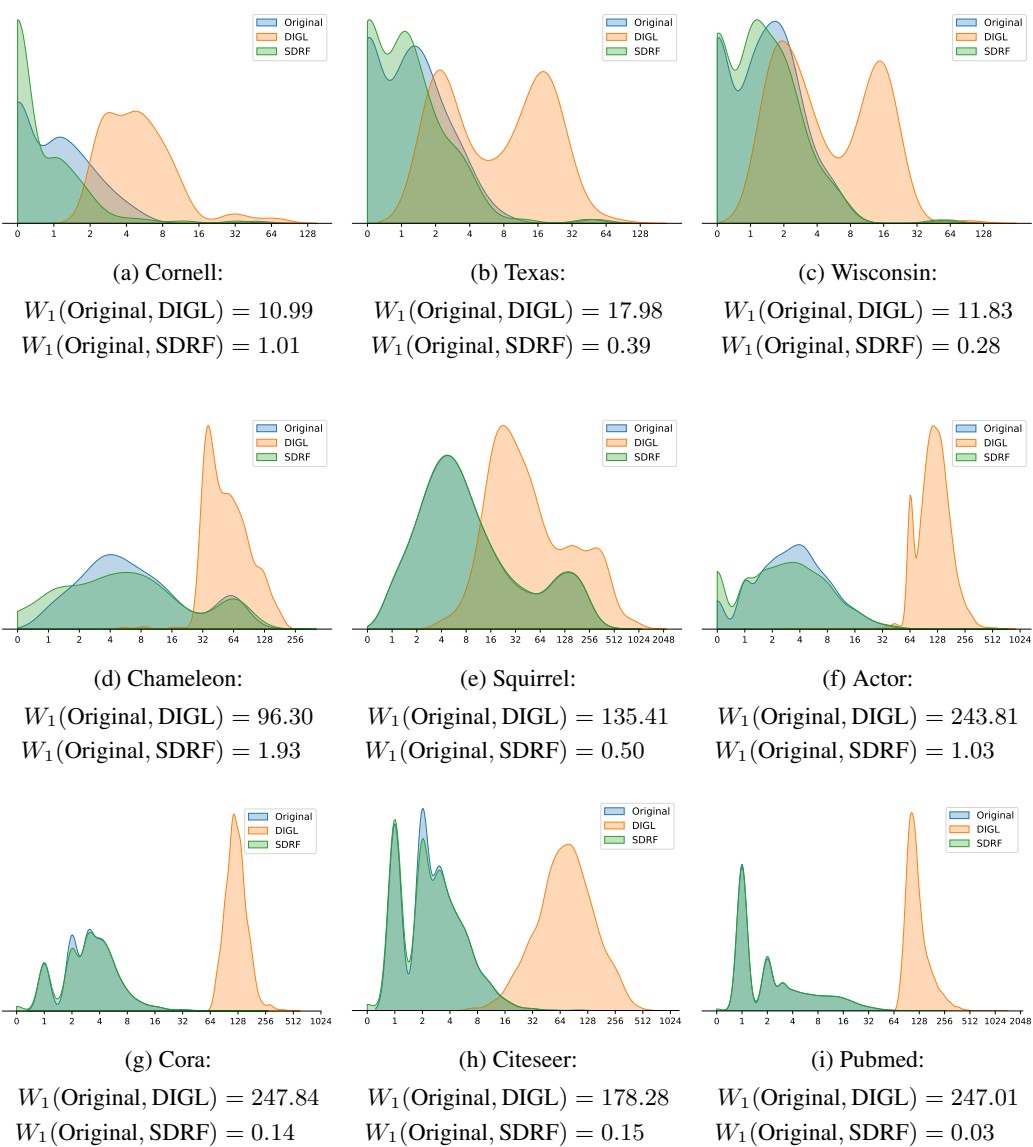

Figure 5: Comparing the degree distribution of the original graphs to the preprocessed version. The x-axis is node degree in $\log_2$ scale, and the plots are a kernel density estimate of the degree distribution. In the captions we see the Wasserstein distance $W_1$ between the original and preprocessed graphs.

## F.3 VISUALIZING CURVATURE AND SENSITIVITY TO FEATURES

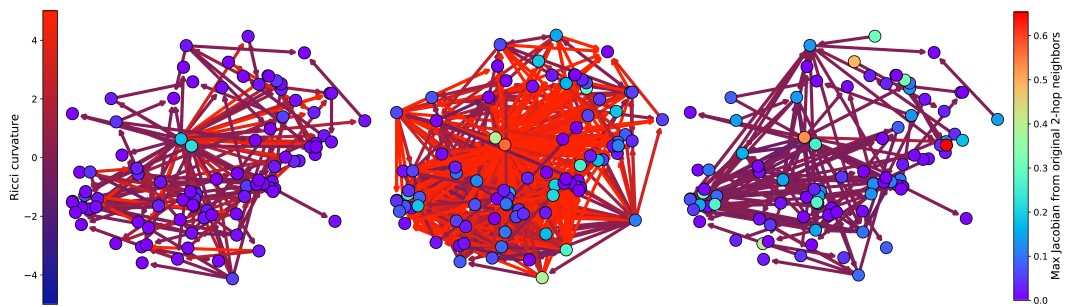

Figure 6: Rewiring of the Cornell graph. Left-to-right: original graph, DIGL, and SDRF rewiring. Edges are colored by curvature; nodes are colored by the maximum absolute entry of a trained 2-layer GCN's Jacobian between the GCN's prediction for that node and the features of the nodes 2 hops away in the original graph. SDRF homogenizes curvature and so lifts the upper bound on the Jacobian from Theorem 4. DIGL also does to an extent, though at the expense of preserving graph topology.

## F.4 HYPERPARAMETERS

Table 4: Hyperparameters for GCN with no preprocessing (None).

| Dataset | Dropout | Hidden depth | Hidden dimension | Learning rate | Weight decay |
|---|---|---|---|---|---|
| Cornell | 0.3060 | 1 | 128 | 0.0082 | 0.1570 |
| Texas | 0.2346 | 1 | 128 | 0.0072 | 0.0037 |
| Wisconsin | 0.2869 | 1 | 64 | 0.0281 | 0.0113 |
| Chameleon | 0.7304 | 3 | 128 | 0.0248 | 0.0936 |
| Squirrel | 0.5974 | 2 | 64 | 0.0136 | 0.1346 |
| Actor | 0.7605 | 1 | 64 | 0.0290 | 0.0619 |
| Cora | 0.4144 | 1 | 64 | 0.0097 | 0.0639 |
| Citeseer | 0.7477 | 1 | 128 | 0.0251 | 0.4577 |
| Pubmed | 0.4013 | 1 | 64 | 0.0095 | 0.0448 |

Table 5: Hyperparameters for GCN with the input graph made undirected (Undirected).

| Dataset | Dropout | Hidden depth | Hidden dimension | Learning rate | Weight decay |
|---|---|---|---|---|---|
| Cornell | 0.6910 | 1 | 64 | 0.0185 | 0.0285 |
| Texas | 0.2665 | 1 | 128 | 0.0069 | 0.0035 |
| Wisconsin | 0.2893 | 2 | 128 | 0.0142 | 0.0001 |
| Chameleon | 0.4657 | 3 | 64 | 0.0189 | 0.0423 |
| Squirrel | 0.5944 | 2 | 64 | 0.0081 | 0.0309 |
| Actor | 0.6626 | 2 | 64 | 0.0195 | 0.0219 |

Table 6: Hyperparameters for GCN with the last layer made fully connected (+FA from Alon & Yahav (2021)).

| Dataset | Dropout | Hidden depth | Hidden dimension | Learning rate | Weight decay |
|---|---|---|---|---|---|
| Cornell | 0.2643 | 1 | 128 | 0.0216 | 0.0760 |
| Texas | 0.2207 | 1 | 128 | 0.0102 | 0.4450 |
| Wisconsin | 0.2613 | 3 | 64 | 0.0057 | 0.0131 |
| Chameleon | 0.7783 | 3 | 64 | 0.0156 | 0.0108 |
| Squirrel | 0.3654 | 3 | 64 | 0.0077 | 0.1922 |
| Actor | 0.3824 | 1 | 64 | 0.0165 | 0.1168 |
| Cora | 0.7840 | 2 | 128 | 0.0149 | 0.1429 |
| Citeseer | 0.5460 | 2 | 64 | 0.0066 | 0.0758 |
| Pubmed | 0.3376 | 2 | 128 | 0.0204 | 0.0215 |

Table 7: Hyperparameters for GCN with DIGL preprocessing, or Graph Diffusion Convolution with PPR kernel (DIGL). Descriptions for $\alpha$, $k$ and $\epsilon$ can be found in Klicpera et al. (2019).

| Dataset | Dropout | Hidden depth | Hidden dimension | Learning rate | Weight decay | $\alpha$ | $k$ | $\epsilon$ |
|---|---|---|---|---|---|---|---|---|
| Cornell | 0.6294 | 1 | 64 | 0.0134 | 0.0258 | 0.1795 | 64 | - |
| Texas | 0.2382 | 2 | 128 | 0.0063 | 0.0153 | 0.0206 | 32 | - |
| Wisconsin | 0.2941 | 1 | 128 | 0.0083 | 0.0226 | 0.1246 | - | 0.0001 |
| Chameleon | 0.4191 | 1 | 128 | 0.0052 | 0.0001 | 0.0244 | 64 | - |
| Squirrel | 0.6844 | 1 | 128 | 0.0056 | 0.4537 | 0.0395 | 32 | - |
| Actor | 0.7820 | 1 | 64 | 0.0170 | 0.0102 | 0.1584 | 128 | - |
| Cora | 0.3315 | 1 | 64 | 0.0284 | 0.0572 | 0.0773 | 128 | - |
| Citeseer | 0.5561 | 1 | 64 | 0.0094 | 0.5013 | 0.1076 | - | 0.0008 |
| Pubmed | 0.4915 | 2 | 128 | 0.0057 | 0.0597 | 0.1155 | 128 | - |

Table 8: Hyperparameters for GCN with DIGL preprocessing followed by symmetrizing the graph diffusion matrix (DIGL + Undirected).

| Dataset | Dropout | Hidden depth | Hidden dimension | Learning rate | Weight decay | $\alpha$ | $k$ | $\epsilon$ |
|---|---|---|---|---|---|---|---|---|
| Cornell | 0.6294 | 1 | 64 | 0.0134 | 0.0258 | 0.1795 | 64 | - |
| Texas | 0.2382 | 2 | 128 | 0.0063 | 0.0153 | 0.0206 | 32 | - |
| Wisconsin | 0.2941 | 1 | 128 | 0.0083 | 0.0226 | 0.1246 | - | 0.0001 |
| Chameleon | 0.4191 | 1 | 128 | 0.0052 | 0.0001 | 0.0244 | 64 | - |
| Squirrel | 0.7094 | 1 | 64 | 0.0172 | 0.0192 | 0.1610 | 64 | - |
| Actor | 0.4012 | 1 | 64 | 0.0161 | 0.0141 | 0.0706 | - | 0.0016 |
| Cora | 0.3315 | 1 | 64 | 0.0284 | 0.0572 | 0.0773 | 128 | - |
| Citeseer | 0.5561 | 1 | 64 | 0.0094 | 0.5013 | 0.1076 | - | 0.0008 |
| Pubmed | 0.4915 | 2 | 128 | 0.0057 | 0.0597 | 0.1155 | 128 | - |

Table 9: Hyperparameters for GCN with SDRF preprocessing (SDRF). Max iterations, $\tau$ and $C^+$ are the SDRF parameters described in Algorithm 1.

| Dataset | Dropout | Hidden depth | Hidden dimension | Learning rate | Weight decay | Max iterations | $\tau$ | $C^+$ |
|---|---|---|---|---|---|---|---|---|
| Cornell | 0.2411 | 1 | 128 | 0.0172 | 0.0125 | 135 | 130 | 0.25 |
| Texas | 0.5954 | 1 | 128 | 0.0278 | 0.0623 | 47 | 172 | 2.25 |
| Wisconsin | 0.6033 | 1 | 128 | 0.0295 | 0.1920 | 27 | 32 | 0.5 |
| Chameleon | 0.7265 | 1 | 128 | 0.0180 | 0.2101 | 832 | 77 | 3.35 |
| Squirrel | 0.7401 | 2 | 16 | 0.0189 | 0.2255 | 6157 | 178 | 0.5 |
| Actor | 0.6886 | 1 | 128 | 0.0095 | 0.0727 | 1010 | 69 | 1.22 |
| Cora | 0.3396 | 1 | 128 | 0.0244 | 0.1076 | 100 | 163 | 0.95 |
| Citeseer | 0.4103 | 1 | 64 | 0.0199 | 0.4551 | 84 | 180 | 0.22 |
| Pubmed | 0.3749 | 3 | 128 | 0.0112 | 0.0138 | 166 | 115 | 14.43 |

Table 10: Hyperparameters for GCN with the input graph made undirected followed by SDRF preprocessing (SDRF + Undirected).

| Dataset | Dropout | Hidden depth | Hidden dimension | Learning rate | Weight decay | Max iterations | $\tau$ | $C^+$ |
|---|---|---|---|---|---|---|---|---|
| Cornell | 0.2911 | 1 | 128 | 0.0056 | 0.0336 | 126 | 145 | 0.88 |
| Texas | 0.2160 | 1 | 64 | 0.0229 | 0.0137 | 89 | 22 | 1.64 |
| Wisconsin | 0.2452 | 1 | 64 | 0.0113 | 0.1559 | 136 | 12 | 7.95 |
| Chameleon | 0.4886 | 1 | 32 | 0.0268 | 0.4056 | 2441 | 252 | 2.84 |
| Squirrel | 0.4249 | 1 | 64 | 0.0295 | 0.1397 | 787 | 43 | 17.19 |
| Actor | 0.6705 | 1 | 128 | 0.0115 | 0.0447 | 1141 | 44 | 11.17 |

## G HARDWARE SPECIFICATIONS

Our experiments were performed on a server with the following specifications:

| Component | Specification |
|---|---|
| Architecture | x86_64 |
| CPU | 40x Intel(R) Xeon(R) Silver 4210R CPU @ 2.40GHz |
| GPU | 4x GeForce RTX 3090 (24268MiB/GPU) |
| RAM | 126GB |
| OS | Ubuntu 20.04.2 LTS |

