# OpenReview forum: "Understanding over-squashing and bottlenecks on graphs via curvature"
_ICLR.cc/2022/Conference — ICLR 2022 Oral_

### Official Review · Reviewer_LDyA · 2021-10-22

**Correctness:** 4
**Technical Novelty And Significance:** 4
**Empirical Novelty And Significance:** 3
**Recommendation:** 8
**Confidence:** 3

**Main Review:**

The paper is well-written and well-organized. Formalizing the concept of over-squashing is interesting and timely, and is of interest to the field. This paper, as is often the case when one delves into novel grounds, makes somewhat arbitrary choices in order to move forward: the definition of over-squashing is somewhat arbitrary, the choice of working with that particular definition of Ricci curvature could be more motivated. Efforts are however made in an interesting direction, and the several discussions around the Cheeger constant gives depth to the paper, and its main messages.

- minor: Appendices F and G are announced at the beginnning of the Appendix section but I did not have them in my pdf document


**Summary Of The Paper:**

In this paper, the authors work on the over-squashing effect that has been recently observed in the GCN literature, namely the effect that, in a message-passing paradigm and for a learning problem that necessitates long-range interaction between distant nodes, the existence of bottlenecks in the graph (for instance an edge e with very large betweenness centrality) will intuitively distort the information that needs to travel from distant nodes and thus hinder the GCN's performance.

One natural direction of research is to add edges (and sometimes remove others in order to keep the complexity under control) to alleviate the bottlenecks. This rewiring process may be done in different ways and the authors suggest one particular algorithm that they compare with state-of-the-art.

In particular, the contributions are:
- a precise (even if somewhat arbitrary) definition of an over-squashing measure between two nodes $i$ and $s$ that is simply the Jacobian of the node representation at $i$ with respect to the entry $x_s$. The smaller this measure, the less $i$ "feels" $s$, the larger the over-squashing effect. The goal of the paper is to find a way to understand how to alleviate the bottlenecks responsible for this effect

- to this end, the authors suggest a new definition of Ricci-like curvature defined over the edges of the graphs. This new definition has one main property which is theorem 3: it lower bounds the Ollivier curvature. This then enables to obtain Cor. 4, and finally the main result Thm 5, that states, in a nutshell, that negatively curved edges are the ones causing bottlenecks.

- a concrete rewiring algorithm is suggested, in which two steps are repeated until convergence or max iteration is reached: a first stochastic step is performed where an edge is added to alleviate the edge that has minimal curvature, before a deterministic step where the edge with maximal curvature (typically those in cliques) is removed if it is larger than a threshold

- a tentative analysis, in the form of Thm 8, as well as experiments show the authors' method performs as well or outperforms other rewiring methods (they mainly compare with DIGL) while keeping the number of total edges after rewiring under control

**Summary Of The Review:**

This is novel work on a seemingly important effect hindering GCN performance in general. Giving a formal definition of over-squashing and providing an analysis based on curvature is novel work, and potentially very useful to the field.

---

> ### Author Response · Authors · 2021-11-17
> **Reply #1 to Official Review of Paper1302 by Reviewer LDyA**
>
> We thank the reviewer for their analysis and for acknowledging both the relevance of the over-squashing issue in the GNN dynamics and the need for a formalization of such problem. We present below an answer to what the reviewer perceived as some level of arbitrariness in the notions we introduced and discussed.
>
> Concerning the somewhat arbitrary nature of definitions, we agree with the reviewer that when proposing new formalizations of empirical phenomena a level of arbitrariness is needed. Still, we wish to motivate a bit further our choices. Concerning the notion of over-squashing, rather than proposing a specific definition we suggested the Jacobian of node representations with respect to input features as a key object in assessing the dependence (or influence score as also noted in Xu et al., 2018). This is arguably the right mathematical quantity to monitor across an architecture to verify how information (messages) are propagating. The notion of curvature we introduced is again not the only working one. We proposed this definition because it both relates nicely to the well-known Ollivier curvature as derived in Theorem 2 and it allows us to control the neighbourhood of an edge precisely hence arriving to the main result in Theorem 4. Other curvature notions may be suitable to control the graph topology and investigate the existence of bottlenecks.
>
> > minor: Appendices F and G are announced at the beginnning of the Appendix section but I did not have them in my pdf document.
>
> Thank you for pointing out the Appendix contents - the reference to Appendix F has been removed, and Appendix G (now Appendix F) describing the hardware specifications has been included in the PDF.

---

> > ### Comment · Reviewer_LDyA · 2021-11-20
> > **Thank you for your reply**
> >
> > Dear authors,
> > good work with this paper.
> > Best;

---

> > > ### Author Response · Authors · 2021-11-22
> > > **Reply #2 to Official Review of Paper1302 by Reviewer LDyA**
> > >
> > > Thank you! We've also now expanded our experiments and analysis to better support the theory in our paper, the details of which are laid out in General Comment #2.

---

### Official Review · Reviewer_kUJT · 2021-10-25

**Correctness:** 4
**Technical Novelty And Significance:** 4
**Empirical Novelty And Significance:** 2
**Recommendation:** 10
**Confidence:** 4

**Main Review:**

## Strengths
* The paper provides good intuitions to over-squashing. This intuition is visualized in Figure 1 that gives a good intuition of over-squashing and of the proposed SDRF solution.
* Further, the paper provides a good connection to graph curvatures that allows to measure and analyze over-squashing.
* The paper provides a thorough analysis using the proposed notations and connects it to further geometric theoretical and classical ideas.
* The paper provides a good conceptual comparison to random-walk-based rewiring and whether this can address the GNN bottleneck, and also an interesting discussion about the likelihood of GNN bottlenecks in homophilic and heterophilic datasets.
* The evaluation shows that the proposed SDRF method outperforms the DIGL baseline without a significant increase in the amount of computation compared to the original graph structure, while the DIGL baseline adds an order of magnitude more edges, which increases the computation cost.

## Weaknesses

* Generality - I am not sure how do these results generalize beyond Graph Convolutional Networks (GCN). The use of GCN's augmented normalized adjacency matrix is assumed at the beginning of the paper, and this assumption is never questioned.
Are Lemma 1 and Corollary 2 correct only for GCN, where it is common to use the augmented normalized adjacency matrix? Specifically, this matrix has $D^{-1}$ which decays messages over paths. What happens if nodes are summed (as in GIN)? How does the signal then decay with the number of layers? Intuitively, I can understand that summing an exponentially growing number of nodes in a single vector cannot compress all the information after some compression steps (that is, GNN layers). But how is this expressed in the curvature theory? If the analysis in the paper refers only to GCN and GNNs which perform averaging, please mention this explicitly.

* Writing - as is, the paper can be perfectly understood only by audiences that are very familiar with all related work. The paper can be significantly improved by giving more background, explaining equations more intuitively, and avoiding unneeded citations which confuse the reader. For example, the paper uses the names such as "Ricci", "Poincare", "Ollivier", and "Forman" frequently. These references can confuse audiences who are not familiar with all terms and papers. Specifically, calling the main method "Balanced Forman", gives the reader the feeling that they would not really understand the proposed method without understanding the original Forman first. I believe that this is not the case, and that the paper *could* be standalone.
While the authors show an impressive understanding and familiarity with the related and classical work, I advise the authors to avoid this "name-dropping", in favor of better readability and accessibility to a broader audience.
I even think that some parts of the paper can be moved to the appendix, in favor of the readability of the remaining parts. For example, I am not sure that Corollary 2 is new (see below), and I'm not sure whether this is directly related to the point the paper tries to make, or whether is it a side-result.
See more specific details below.

* The Evaluation could also be strengthened by addressing more datasets and more GNN types.
See more details below.

* Reproducibility - please provide more training details, hyperparameters tuned and ranges, and the final selected hyperparameter values for the experiments in Section 5. A table of dataset statistics (in the appendix) can also help, before and after applying every type of preprocessing method (similarly to Table 3, possibly with absolute numbers as well).


## Writing

* Section 2.1: what is "$(i,j) \in E$ if $i \sim j$"? What does $i\sim j$ mean? Does it simply mean that there is an edge between them? If it only means that there is an edge between them, why is this a different notation than $(i,j) \in E$?

* What is $\hat{a}$ in Equation (1)? I cannot find its definition. Is it $\hat{A}$? It is unclear from the text.

* Section 2.1:
>"$d_{G}$ is the standard minimum-walk (geodesic) distance on the graph"

Is it simply the **shortest-path**? If so, I suggest using this term which is much more common to the broader audience. If not, the term "minimum-walk (geodesic) distance" should be better explained.

* Section 2.1:
> "the node features and representations are assumed to be scalar from now on"

that is, $p_0 = 1$ and all $p_l$ equal to 1?

* Corollary 2 - I think that the "Jumping Knowledge" paper (Xu et al., 2018) had already recognized this role of self-loops. What is the difference between this corollary and Theorem 1 in Xu et al. (2018)? Also, as the role of self-loops is not directly connected to the main contributions of this paper, I suggest considering moving it to the appendix, or removing it if it was already recognized by Xu et al, in favor of more space for explanations of other parts.

* Section 3, definition of $\lambda_{max}$ - what does "traversing the same node" mean? the same $k$ node from the previous definition of 4-cycles? A formal definition of $\lambda_{max}$ will be helpful.

* Section 3, the example describing Figure 3: I cannot understand the examples because in (ii), $\sharp_{\square}^{i}$ is defined for a specific edge $\sharp_{\square}^{i}(i,j)$, and here it appears as $\sharp_{\square}^{0}$ and $\sharp_{\square}^{1}$

Also, why is $\sharp_{\square}^{1}={5}$, and not also the node 0 is included, as it also creates a 4-cycle (1-0-3-5)?

* Corollary 4: what is "the volume of the geodesic balls", and what does growing "polynomially" exactly mean? How is the volume measured, and what does it grow over? (what is the "x axis"?)

* Theorem 5 - it would help if the authors could clarify this theorem, explain its meaning in words and intuitively. I am not sure I understand why every detail there is necessary, and what are its implications.

* Cheeger constant (Equations (5)+(6)) - I did not really understand the notations in Eq. (5), what is the meaning of Equation (6), what is its significance?

##  Evaluation:
1. It would really improve the evaluation if the authors could experiment with more GNN types and more benchmarks. Specifically, if the authors could take the datasets and exact settings used in Alon & Yahav (2020) and show how the proposed SDRF method improves the results there. The reason that this would be helpful is that Alon & Yahav showed that these datasets already suffer from over-squashing, by taking existing source code of other papers and improving their results. It would be interesting to compare SDRF and the (computationally expensive) solution of Alon & Yahav in terms of the tradeoff between accuracy and computation cost (the overall number of added edges).

2. DIGL is compared to SDRF as the main baseline. However, I couldn't find what exactly does DIGL do. The paragraph about random-walk-based rewiring on page 7 does explain the general idea, but it is unclear whether this is explaining directly about DIGL, or in general about PageRank-style approaches?

## Additional questions to authors
* According to this analysis, in the authors' opinion, are over-squashing and over-smoothing the same thing or not? What is the difference between them? (The answer to this question will **not** affect the rating negatively, as it is obviously out of the scope of this paper)
If the authors have an insightful explanation, I recommend including it in the body of the paper.



### Other minor comments:
* The paragraph about Discrete curvatures on graphs on page 4 is completely unclear for the audience who are unfamiliar with the described curvatures. I suggest delaying this to later or even moving to an appendix in favor of more text that will help clarify the rest of the paper.

* Theorem 5(i) writes $\ell \in [0, L-1]$. For clarity, I suggest being consistent which previous notations, for example as Lemma 1 which denotes $0 \leq \ell \leq ...$.


**Summary Of The Paper:**

 The paper provides an analytical explanation for the over-squashing and GNN bottleneck phenomena. While Alon & Yahav (2020) demonstrated the over-squashing phenomenon empirically and provided mainly intuitions as to why it happens, this paper performs a deeper analysis and connects over-squashing to combinatorial curvatures. The paper shows that negatively curved edges are responsible for over-squashing. This is important because it allows to measure over-squashing, and pinpoint specific edges that are responsible for it in a given graph.
Further, the paper proposes a method (called SDRF) to re-wire the graph based on these insights and shows empirically how this method alleviates over-squashing.


**Summary Of The Review:**

Despite my many comments, I think that this is an important paper, which deepens our understanding of the over-squashing phenomenon from a geometric perspective.
While many recent GNN papers introduce new application domains, new features, and new GNN architectures, this paper improves our understanding of the foundations of existing GNNs and their limitations, which is even more important.

I will increase my rating if the authors would answer all questions in the "Writing" section above, in the "Additional Questions" section above, and provide more evaluation datasets (as detailed in the "Evaluation" section above).

===== Update =====

I have increased my rating to 10, good work.

Note that Table 2 is now too wide and goes beyond the page's borders.

---

> ### Author Response · Authors · 2021-11-17
> **Reply #1 (Part 1) to Official Review of Paper1302 by Reviewer kUJT**
>
> We thank the reviewer for their thorough analysis and helpful and constructive feedback. We address below each point raised by the reviewer in order while also referring to the general comment above where we have listed modifications we have already made to the submission. As you have given us a detailed response we also want to respond thoroughly, and so our reply is broken into parts due to character limits.
>
> > Generality - If the analysis in the paper refers only to GCN and GNNs which perform averaging, please mention this explicitly.
>
> Although we have presented the sensitivity analysis for the case of averaging aggregations with normalized adjacency, one can also rephrase over-squashing results in terms of the Jacobian for GNNs where features are simply summed in the aggregation step (i.e. the adjacency is not normalized). Consistently with Lemma 1, we restrict to the setting where features and node representations at each layer are scalars to make the discussion simpler. In line with the _Jumping knowledge_ paper (Xu et al., 2018) we consider a GNN-model of the form
>
> $ h_{i}^{(\ell + 1)} = \text{ReLU}\left(\\sum_{j\in \tilde{N_i}}h_{j}^{(\ell)}w_{\ell}\right). $
>
> The augmented neighbourhood $ \tilde{N_i} $ is defined as $N_i\cup \{i\}$. Differently from the setting of Theorem 1 in the _Jumping knowledge_ paper, the aggregation here is not an average but a simple sum. Let us now take nodes $i$ and $s$ such that $s\in S_{r+1}(i)$ as in the statement of Lemma 1 in our submission. In this case, instead of simply considering the quantity $\lvert \partial h_{i}^{(r+1)}/\partial x_{s}\rvert$ we normalize the Jacobian entries, obtaining what is referred to as _influence score_ in (Xu et al., 2018):
>
> $ J_{r+1}(i,s) := \frac{\left\vert \frac{\partial h_{i}^{(r+1)}}{\partial x_{s}} \right\vert}{\sum_{k}\left\vert \frac{\partial h_{i}^{(r+1)}}{\partial x_{k}} \right\vert} $
>
> This now represents a relative importance of feature $x_{s}$ on the representation of node $i$ at layer $r+1$. If - similarly to Theorem 1 in (Xu et al., 2018) - we assume that all paths in the computational graph of the model are activated with the same probability, then we obtain that on average
>
> $ J_{r+1}(i,s) = \frac{ \tilde{A}\_{is}^{r+1} }{ \sum\_k\tilde{A}^{r+1}_{ik} } \leq \frac{\tilde{A}^{r+1}\_{is}}{\text{Vol}(B\_{r+1}(i))} $
>
> where $\tilde{A} = A + I$ and $\text{vol}(S) = \sum_{j\in S}d_{j}$. In particular, we again find that if for example we have a tree structure, then the right hand side decays exponentially as $2^{-(r+1)}$.
>
> We have added this comment in Appendix A.
>
> > Writing - as is, the paper can be perfectly understood only by audiences that are very familiar with all related work. (...) For example, the paper uses the names such as "Ricci", "Poincare", "Ollivier", and "Forman" frequently. (...) Specifically, calling the main method "Balanced Forman", gives the reader the feeling that they would not really understand the proposed method without understanding the original Forman first. (...) some parts of the paper can be moved to the appendix (...) For example (...) Corollary 2.
>
> We provide here a general answer while referring below for more detailed comments. We have simplified the discussion in Section 3 about curvature in the smooth and discrete settings. Concerning the name of the curvature, we think that including Forman in the name is justified: this in line with other works where combinatorial notions of curvature on graphs usually contain the name Forman (such as Augmented Forman), which is known in the field (for example see _Comparative analysis of two discretizations of Ricci curvature for complex networks_, Samal et al. 2018). While space constraints prevent us from presenting the main curvature candidates on graphs in the main text which give some background to the name, we refer to Appendix C for a thorough review.

---

> > ### Author Response · Authors · 2021-11-17
> > **Reply #1 (Part 2) to Official Review of Paper1302 by Reviewer kUJT**
> >
> > ### Writing
> > * $(i,j)\in E$ and $i\sim j$ are equivalent notations to denote edges. The second one is just a shorthand and is preferred in equations.
> >
> > * Yes $\hat{a}$ in equation(1) should be $\hat{A}$ and we have changed it.
> >
> > * Yes $d_{G}$ is the shortest path. We have changed the name as suggested by the reviewer.
> >
> > * "the node features and representations are assumed to be scalar from now on" - yes, this means that $p_{0} = p_{\ell} = 1$.
> >
> > * We agree with the reviewer that the role of self-loops as pointed out in Corollary 2 is not specifically relevant to the message of the paper and we have hence moved the Corollary to the Appendix. Concerning the differences between Corollary 2 and Theorem 1 in (Xu et al., 2018) we believe that the first result is not exactly contained in the second one. First, the role of self-loops seems to be only implicitly relied on in the _Jumping knowledge_ paper where the authors associate them to lazy random-walks. This has not been made explicit in the statement of Theorem 1 which also proposes an expectation type of result based on the ReLU activation function. On the other hand, Corollary 2 shows how for a graph neural network without self-loops - irrespective of the adopted non-linear activation and of the lazy random-walk interpretation - a node $i$ representation at layer $\ell$ only depends on features reachable by walks of length _exactly_ $\ell$. As far as we can see, this does not seem to be easily derivable from the analysis in (Xu et al., 2018). We have also added a Remark at pag. 14 in the Appendix pointing out to (Xu et al., 2018) about the connection between self-loops and lazy random walks.
> >
> > * The precise definition of $\lambda_{\text{max}}$ can be found in Definition 4 in the Appendix, pag. 15. We now refer to it explicitly in the main text. Note that to avoid confusion with standard notations for eigenvalues we have renamed it to be $\gamma_{\text{max}}$.
> >
> > * In the discussion surrounding Figure 3 we have used a shorthand $\sharp_{\square}^{0}$ for $\sharp_{\square}^{0}(0,1)$. We have changed the shorthand back to the usual notation to avoid confusion.
> >
> > * The node $0$ in the computation of $\sharp_{\square}^{1}(0,1)$ is not included because as specified in the definition of $\sharp_{\square}^{i}(i,j)$ we never consider the node $j$ (this appears as the requirement $k \neq j$ in the definition).
> >
> > * In Corollary 4 the volume of geodesic balls refers to the quantity $\lvert B_{r}(i)\rvert$, for a given node $i$ and radius $r$, i.e. it is the number of nodes at distance smaller or equal than $r$ from node $i$. The volume of geodesic balls _grows polynomially_ if
> > $  \lvert B(i,r)\rvert \leq P(r), $
> > for any node $i\in V$, with $P$ a polynomial in $r$. We have added this formula in the statement of Corollary 4 to make the result more accessible.
> >
> > * We comment here about the details of Theorem 5 (now Theorem 4 in the revised submission). We have added further explanation in the paper as well. First about the assumptions: the requirement in (i) is just to control the message passing functions - in the linear case we would have the norm of the weight matrices. Assumption (ii) instead means that the curvature of the given edge is negative enough when compared to the degrees of the endpoint nodes (recall that by our definition $\text{Ric}(i,j) > -2$). The further condition on $\gamma_{\text{max}}$ is meant to avoid pathological cases where we have a large number of degenerate 4-cycles passing through the same three nodes. If these conditions are satisfied, then we are able to prove that negatively curved edges are those creating bottleneck regarded as those graph edges where the propagation of information is negatively affected. More precisely, we show that there exist a large number of nodes $k$ such that GNNs - on average - struggle to propagate messages from $i$ to $k$ in two layers despite these nodes $k$ being at distance $2$ from $i$. This is measured explicitly by the Jacobian (or influence score).
> >
> >
> > * The notations in equation (5) are the standard ones in spectral graph theory. First, one defines a constant $h_{S}$ for each subset $S\subset V$ to be the ratio between the number of edges from $S$ to its complement and the smallest volume among $S$ and its complement. The Cheeger constant is then defined as the smallest such $h_{S}$ over all possible subsets $S$. The significance of equation (6) - both for manifolds and graphs - mainly boils down to the possibility of relating spectral properties of an operator (the Laplacian) to topological features of the space (in this case the Cheeger constant).

---

> > > ### Author Response · Authors · 2021-11-17
> > > **Reply #1 (Part 3 of 3) to Official Review of Paper1302 by Reviewer kUJT**
> > >
> > > ### Evaluation
> > >
> > > > It would really improve the evaluation if the authors could experiment with more GNN types and more benchmarks. Specifically, if the authors could take the datasets and exact settings used in Alon & Yahav (2020) and show how the proposed SDRF method improves the results there.
> > >
> > > We agree that strengthening the experiment section of this work would increase the support for the theory presented. To this end we are performing experiments on datasets from the wider standard suite such as Citeseer, Pubmed, Chameleon and Squirrel. We will also be including the +FA method proposed in Alon & Yahav (2020) as an additional benchmark. If time allows we will reproduce the experiments with GAT and GIN to increase variety in GNN type. The experiments in Alon & Yahav (2020) are something we would like to reproduce with SDRF in future, but due to the difference in codebases it is unlikely we will have time to do this before the end of the discussion period.
> > >
> > > > It would be interesting to compare SDRF and the (computationally expensive) solution of Alon & Yahav in terms of the tradeoff between accuracy and computation cost (the overall number of added edges).
> > >
> > > We agree, and we will include the +FA benchmark in our comparison of the methods' topological effects on the graph (including proportion of added edges and the other graph metrics to be added).
> > >
> > > > DIGL is compared to SDRF as the main baseline. However, I couldn't find what exactly does DIGL do. The paragraph about random-walk-based rewiring on page 7 does explain the general idea, but it is unclear whether this is explaining directly about DIGL, or in general about PageRank-style approaches?
> > >
> > > DIGL is a pre-processing step proposed in Klicpera et al. (2019) in which the given adjacency $A$ is replaced with a new adjacency $A'$ obtained by some diffusion process on $A$. One of the main diffusion processes analysed in their paper is the PPR method, where $A'$ is obtained by sparsifying what we refer to as $R_{\alpha}$ in our submission. In Theorem 6 we generally show that the PPR approach is generally not suited to address structural features of the graph like the bottleneck by showing that in general if we take $A' = R_{\alpha}$, it is not possible to improve the Cheeger-constant of the new rewired graph arbitrarily well with respect to the Cheeger constant of the input graph (in contrast to a curvature-based approach). The result does not hold for the approach in DIGL only, but also applies to any method relying on the PPR matrix $R_{\alpha}$ derived from a given adjacency $A$. We refer to Proposition 17 and Remark 18 in Appendix E for results that are more tailored to the actual strategy adopted in DIGL since there we also take into account the effect of the sparsification.
> > >
> > > ### Additional questions to the authors
> > > In our opinion over-smoothing and over-squashing are in principle different phenomena. The first occurs on diffusion-based GNN architectures when the depth becomes very large and is independent of the graph-topology and the learning task. Over-squashing instead occurs on any message-passing model depending on the graph-topology (i.e. do we have a bottleneck?) and the learning task (do we have long-range dependencies?) but is independent of the depth. Sometimes there might be a correlation among the two: for example, if we have long-range dependencies we might need many layers hence promoting over-smoothing. In general though, we believe these two problems should be dealt with as distinct issues.
> > >
> > > ### Other minor comments
> > > * We have modified the paragraph about discrete curvature streamlining the discussion further.
> > > * Notations have been updated as suggested by the reviewer in the statement of Theorem 4.

---

> > > > ### Comment · Reviewer_kUJT · 2021-11-18
> > > > **Thank you for your response**
> > > >
> > > > Thank you for your response.
> > > >
> > > > I think that the paper is accessible to a broader audience now!
> > > >
> > > > 1. I suggest including the description of DIGL more explicitly in Section 5, as it is the main baseline.
> > > > 2. Thank you for the explanation about over-smoothing and over-squashing, I agree with the authors' opinions. What kinds of GNNs do you consider as "diffusion-based GNN architectures", in contrast with "any message-passing model"? If the authors find this relevant, consider including this discussion in the paper or the appendix as well.
> > > > 3. Minor comment: when looking up some of the referenced papers, I noticed that some bib entries refer to the arxiv version where a more updated version exists which includes more information like the discussion, reviews, supplementary material, etc. This is minor, but since the paper depends on previous literature so heavily, I suggest going through the bibliography and trying to replace arxiv entries, to make it easier for the readers to lookup papers. For example Kipf & Welling, Alon & Yahav, Liu et al., Klicpera et al., Defferrard et al.,  Velickoviˇc et al.,.
> > > >
> > > > I am looking forward to the additional experimental results.

---

> > > > > ### Author Response · Authors · 2021-11-18
> > > > > **Minor modifications added**
> > > > >
> > > > > Thank you for your prompt reply.
> > > > >
> > > > > 1. We have added a more explicit description of DIGL along the lines of our comment in the main text.
> > > > >
> > > > > 2. Although this is an interesting point, we think it might shift the focus of the paper. For the sake of completeness, we limit ourselves to mentioning the following here. In a nutshell, one can find message passing functions that might not promote averaging/diffusion (constant signals in the limit) and hence in principle avoid over-smoothing. However, for such functions, over-squashing might still occur whenever topological bottlenecks and long-range dependencies exist.
> > > > >
> > > > > 3. We thank the reviewer for pointing this out. We have updated the references accordingly.

---

> > > > > > ### Comment · Reviewer_kUJT · 2021-11-18
> > > > > > **Thanks**
> > > > > >
> > > > > > 1. Thank you
> > > > > > 2. I agree that it might shift the focus of the paper. Thank you for the explanation.

---

> > > > > > > ### Author Response · Authors · 2021-11-22
> > > > > > > **Reply #3 to Official Review of Paper1302 by Reviewer kUJT**
> > > > > > >
> > > > > > > Our changes in this revision are described in General Comment #2, but as you said you were looking forward to them we'd like to highlight the additional experimental results. We feel that the added inclusion of Chameleon, Squirrel, Actor, Citeseer and Pubmed as datasets, +FA as a baseline method, and the new analysis on degree distributions make for a stronger paper. We again thank you for your feedback and for engaging in helpful discussion.

---

### Official Review · Reviewer_idLR · 2021-11-02

**Correctness:** 3
**Technical Novelty And Significance:** 3
**Empirical Novelty And Significance:** 3
**Recommendation:** 8
**Confidence:** 3

**Main Review:**

Detailed comments:
- A critical aspect of rewiring is its effect on the structure and topological properties of the underlying graph. This is not captured by just analyzing the number of added/ removed edges across the graph. I think that it would be more useful to analyze graph characteristics (such as the node degree distribution) or to measure the distance of the original and the rewired graph globally, e.g., with a transportation distance.
- Can you comment on the cost of curvature-based rewiring vs. random walk-based rewiring? If curvature-based rewiring is less efficient than random walk-based rewiring, is this (in your experiments) mitigated by a reduced cost in the downstream task (due to the smaller number of edges).
- In section 4 you briefly remark that curvature-based rewiring may reduce over-smoothing. I think it could be interesting to expand on that.


**Summary Of The Paper:**

The authors propose a new graph rewiring approach that utilizes a discrete notion of Ricci curvature to mitigate over-squashing. This is motivated by a link between negatively curved edges and graph bottlenecks. The paper has a theoretical focus, but also provides a set of validation experiments to demonstrate the proposed approach.

**Summary Of The Review:**

I found the paper very interesting in that it analyzes the over-squashing problems through a new, geometric lens. The theoretical motivation for using curvature-based tools as opposed to random-walk-based rewiring is well-done. My main concern is that the authors’ notion of graph structure preservation is not convincing to me. The experiments could be more comprehensive. Please see detailed comments.

---

> ### Author Response · Authors · 2021-11-17
> **Reply #1 to Official Review of Paper1302 by Reviewer idLR**
>
> We thank the reviewer for their comments, for their positive feedback on our geometric approach to the over-squashing issue and for their suggestions on how to improve the discussion about preservation of graph topological properties when comparing different rewiring methods. We have addressed below the comments in order.
>
> > A critical aspect of rewiring is its effect on the structure and topological properties of the underlying graph. This is not captured by just analyzing the number of added/ removed edges across the graph. I think that it would be more useful to analyze graph characteristics (such as the node degree distribution) or to measure the distance of the original and the rewired graph globally, e.g., with a transportation distance.
>
> Since any graph-rewiring method inevitably modifies the graph structure to reduce some negative effect or promote beneficial ones, the graph-edit distance, albeit elementary, offers a simple way of comparing the input graph with the rewired one. In this regard, any localized rewired method as the one we propose is bound to modify the graph topology only where needed hence allowing us to control the graph-edit distance more easily. However, we agree that more sophisticated methods of assessing the distance between the input graph and the rewired one should also be tested. We first note that as argued at page 7 of our submission, our curvature based method will generally preserve degree-distribution since edges with very large curvature are on average likely to have endpoints with large degree. Namely, we noted how $\text{Ric}(i,j) < -2 + \delta$, for some $\delta \in (0,2)$, implies that either $d_{i}$ or $d_{j}$ must be larger than $2/\delta$. To help measure this we will also include the effect on some summary statistics for the degree distribution.
>
> Investigating transportation distance also seems a good option. We offer an intuitive picture on why curvature-based methods should better preserve transportation distances among graphs. Since our SDRF algorithm works by adding edges among nodes that are at most at distance 2 on the given graph, the matrix of pairwise distances is only going to be modified by 1 for each pair of nodes whose minimal path passed through $i$ and $j$. In other words, since our rewiring steps are local and add edges among nearby nodes, the matrix of distances is affected only in a limited matter. Accordingly, we expect the Gromov-Wasserstein distance between the input graph and the rewired one to be generally smaller for our curvature-based method when compared to random-walk diffusion approaches. If time permits, we will also provide comparisons in our evaluation section for the Gromov-Wasserstein distance.
>
> > Can you comment on the cost of curvature-based rewiring vs. random walk-based rewiring? If curvature-based rewiring is less efficient than random walk-based rewiring, is this (in your experiments) mitigated by a reduced cost in the downstream task (due to the smaller number of edges).
>
> SDRF, our initial example of curvature-based rewiring, often does takes more time than DIGL due to its iterative structure. This being said, as you say we do see that the downstream GNN training is often faster on the datasets preprocessed with SDRF than with DIGL due to the computational cost from the volume of edges often added by DIGL. We will include some plots comparing the preprocessing and training times for the two methods for some of the experimental settings.
>
> > In section 4 you briefly remark that curvature-based rewiring may reduce over-smoothing. I think it could be interesting to expand on that.
>
> We thank the reviewer for pointing this out. We have decided to remove this sentence since expanding on that could take a lot of space and compromise the flow of the paper. First, as we have also commented on in another review, over-smoothing and over-squashing are in principle different phenomena. The first occurs on diffusion-based GNN architectures when the depth becomes very large and is independent of the graph-topology and the learning task. Over-squashing instead occurs on any message-passing model depending on the graph topology (i.e. do we have a bottleneck and so have negatively curved edges?) and the learning task (do we have long-range dependencies?) but is independent of the depth. Sometimes there might be a correlation among the two, which is what we were hinting at in the paper: if we have long-range dependencies we might need many layers hence promoting over-smoothing. A rewiring addressing the bottleneck might in principle also decrease the average radius of dependencies meaning that in practice we might need less layers. In general though, we believe these two problems should be dealt with as distinct issues.

---

### Official Review · Reviewer_Za5a · 2021-11-07

**Correctness:** 4
**Technical Novelty And Significance:** 4
**Empirical Novelty And Significance:** 2
**Recommendation:** 8
**Confidence:** 4

**Main Review:**

Overall the paper is well written, however, the paper improve further to have better readability. It would be helpful to have more intuitive clarity on the use of hyperbolic spaces since many readers may not be familiar with the topic. Throughout the paper, homophilic and hetherophilic nature of graphs are mentioned, however, the exact definitions of them are not clearly defined. Further, in experiments “low-homophily” is mentioned, but there is not measure for low or high homophily. I suggest that the authors provide a proper definition for homophilic/heherophilic as [1,2] and provide numerical measures for datasets used in experiments as in [1]

The main weak point of the paper is experiments. First of all, is it fully-supervised node classification of semi-supervised node classification? The experiments are not substantial in terms of dataset selections, comparisons with baseline methods and obtained accuracy.

a) The selected baseline methods are limited, can the authors use further baselines methods? Can the proposed graph rewiring method be compared with the +FA method in  [1]?

b) To my knowledge Cora is not a homophilic dataset [1]. It would be helpful if more datasets from hetherophilic graphs (Citeseer, Pubmed) are also considered in experiments, which would allow us to  have a better understanding of the over-squashing phenomenon.

c) Cornell, Texas and Wisconsin are small scale datasets. I suggest to experiments with Chameleon or/and Squirrel dataset, which have large number of nodes and dense graphs.

d) Why are accuracies of Cornell, Texas, and Wisconsin low? There are methods that have shown greater accuracy for these datasets [1]. I wonder whether the low accuracy is due to the use of GCN and if a different GNN method is used in with the proposed rewiring method the accuracy would increase. Do authors have any experience with other models?



References
[1] Eli Chien and Jianhao Peng and Pan Li and Olgica Milenkovic, Adaptive Universal Generalized PageRank Graph Neural Network,
International Conference on Learning Representations (2021)

[2] Hongbin Pei, Bingzhe Wei, Kevin Chen-Chuan Chang, Yu Lei, and Bo Yang. Geom-gcn: Geometric graph convolutional networks.  International Conference on Learning Representations (2019)

**Summary Of The Paper:**

This paper analyzes oversquashing in GNN using geometric methods. The paper proposes a novel rewiring method based on negative curvature to construct graphs that are less susceptible to oversquashing. Though the theoretical contribution is strong the empirical evidence is not strong. Overall a good paper.

**Summary Of The Review:**

A good paper with a novel ideas on oversquashing in GNN. The main limitation of the paper is the limited of empirical support.

---

> ### Author Response · Authors · 2021-11-17
> **Reply #1 to Official Review of Paper1302 by Reviewer Za5a**
>
> We thank the reviewer for their feedback, for the positive comment on our theoretical contribution and in particular for pointing us to ways to improve the evaluation section and the discussion about heterophily. We address below the comments in order:
>
> > Overall the paper is well written, however, the paper improve further to have better readability.
>
> We have made changes to the text to reduce the amount of domain-specific terminology without affecting the flow of the paper, as described above in General Comment #1. We have also added further explanation around key results such as Theorem 4.
>
> > Throughout the paper, homophilic and hetherophilic nature of graphs are mentioned, however, the exact definitions of them are not clearly defined. Further, in experiments “low-homophily” is mentioned, but there is not measure for low or high homophily.
>
> We agree. In the revision, we will include the definition of the homophily index introduced in [2] and provide its value on the datasets used in the experimental section as in [1,2].
>
> > ...is it fully-supervised node classification of semi-supervised node classification?
>
> There is some ambiguity here since we have node labels which indicates supervised learning, but as we also use the graph structure that can be viewed as semi-supervised. To keep with the terminology used in Klicpera et al 2019 we will use the term semi-supervised node classification and we have clarified this in the paper.
>
> > a) The selected baseline methods are limited, can the authors use further baselines methods? Can the proposed graph rewiring method be compared with the +FA method in [1]?}
>
> We agree. We will include the +FA method for comparison in the revision.
>
> > b) To my knowledge Cora is not a homophilic dataset [1]
>
> We should perhaps phrase it more accurately. Using the definition of homophily in [1,2], Cora, Citeseer and Pubmed have a homophily index of 0.83, 0.71, and 0.79 respectively compared to 0.11, 0.06 and 0.16 of Cornell, Wisconsin, and Texas. Due to this large difference, we believe in this context it is fair to call Cora, Citeseer and Pubmed 'homophilic' and Cornell, Wisconsin, and Texas 'heterophilic'.
>
> > It would be helpful if more datasets from hetherophilic graphs (Citeseer, Pubmed) are also considered in experiments, which would allow us to have a better understanding of the over-squashing phenomenon. ... c) Cornell, Texas and Wisconsin are small scale datasets. I suggest to experiments with Chameleon or/and Squirrel dataset, which have large number of nodes and dense graphs.
>
> We agree. We will include additional datasets in the revised experimental section.
>
> > d) Why are accuracies of Cornell, Texas, and Wisconsin low? There are methods that have shown greater accuracy for these datasets [1]. I wonder whether the low accuracy is due to the use of GCN and if a different GNN method is used in with the proposed rewiring method the accuracy would increase. Do authors have any experience with other models?
>
> The accuracies shown are are lower compared to better SOTA methods because the experiments are performed with a simple GCN (we used the same PyTorch implementation as in Klicpera et al 2019 available from https://github.com/klicperajo/gdc/blob/master/models.py). We should stress that in our opinion this does not diminish the validity of our theoretical results, since our approach is preprocessing of the graph independent of the GNN architecture used downstream. Since the main contribution of our paper is theoretical, we believe experimenting with different architectures and curvature-based rewiring is important future experimental work. If time allows we will also be including experiments with the GAT and GIN architectures. With regards to [1] specifically, the difference in accuracies may also be influenced by train/val/test splits - [1] uses a 60/20/20 split referred to as "dense", whereas we used splits more like Klicpera et al 2019 referred to in [1] as "sparse".

---

> > ### Comment · Reviewer_Za5a · 2021-11-20
> > **Reply**
> >
> > Thank you for the rebuttal.
> >
> > Please include the clarifications on the experiments in the revised paper. Otherwise, it seems that the authors did not pay careful attention to experiments and show selected datasets that give good performances.
> >
> > I would also like to see updated results (at least for some concerns) with experiments.

---

> > > ### Author Response · Authors · 2021-11-22
> > > **Reply #2 to Official Review of Paper1302 by Reviewer Za5a**
> > >
> > > Thank you for your response. Based on your feedback we have substantially fleshed out the empirical side of our paper, namely:
> > > * Expanding the datasets included to include Chameleon, Squirrel, Actor, Citeseer and Pubmed;
> > > * Added the +FA method as an additional baseline;
> > > * Moved to using "dense" splits for the WebKB datasets as in [1,2] for consistency with existing literature, which has in turn raised accuracies for all experiments on these datasets;
> > > * And more fully described our experiment methodology, along with a measure of homophily from [2], in Section 5 and the new Appendix F.
> > >
> > > Again we thank you for your feedback and we hope that the greater experimental support improves your confidence in our work.
> > >
> > > [1] Eli Chien and Jianhao Peng and Pan Li and Olgica Milenkovic, Adaptive Universal Generalized PageRank Graph Neural Network, International Conference on Learning Representations (2021)
> > >
> > > [2] Hongbin Pei, Bingzhe Wei, Kevin Chen-Chuan Chang, Yu Lei, and Bo Yang. Geom-gcn: Geometric graph convolutional networks. International Conference on Learning Representations (2019)

---

### Author Response · Authors · 2021-11-17
**General response #1 (2021-11-17)**

We would like to thank all the reviewers for taking the time to read our paper thoroughly and for providing insightful and productive comments, both on the theoretical and experimental contents of the paper. Below, we summarize the revisions we have made in response to the reviews.

Edits made in current revision (2021-11-17):
* Added a comment in Appendix A that extends Lemma 1 to include GNN types which do not perform averaging (see our response to Reviewier kUJT);
* Simplified  the  discussion  in  Section  3  about  curvature in the continuous vs discrete settings to improve accessibility for a wider audience;
* Moved Corollary 2 to the Appendix to improve the flow of the paper;
* Corrected $\hat{a}$ to $\hat{A}$ in Equation 1;
* Renamed $d_G$ as shortest-path rather than minimum-walk distance to better follow convention;
* Added a remark at page 14 in the Appendix on the connection between self-loops and lazy random walks noted in (Xu et al., 2018);
* Changed the variable $\lambda_{\text{max}}$ to $\gamma_{\text{max}}$ to avoid confusion with standard notation for eigenvalues, and added an explicit reference in the main text to Definition 4 in page 15 of the Appendix.
* Written $\sharp_{\square}^{1}(0,1)$ instead of the shorthand $\sharp_{\square}^{1}$ in the discussion surrounding Figure 3 to improve readability;
* Added further comments to motivate and explain the significance of our main result in Theorem 4.

To be added over the next few days:
* Experimental results on other datasets including Citeseer, Pubmed, Chameleon, Squirrel, and Actor;
* Addition of the +FA method from Alon et al 2020 (making the last GNN layer fully connected) as a benchmark;
* Further measures of effect on graph structure beyond number of edges added/removed, including effect on degree distribution and a transportation-based metric (we will be looking at Gromov-Wasserstein distance);
* Inclusion of the measure of homophily from [1,2] - the values will be included in the results table in Section 5, and the definition of the homophily index will likely be included in the Appendix due to space constraints and its introduction in other works;
* Inclusion of all hyperparameters used for each preprocessing/model/dataset combination in the Appendix;
* Inclusion of some plots comparing the preprocessing and training times for DIGL and SDRF for some of the experimental settings;
* If time allows, we will reproduce the experiments with GAT and GIN to demonstrate the methods on other GNN types.

Again we thank you for your responses and feedback and we look forward to discussion over the next week.

[1] Eli Chien and Jianhao Peng and Pan Li and Olgica Milenkovic, Adaptive Universal Generalized PageRank Graph Neural Network, International Conference on Learning Representations (2021)

[2] Hongbin Pei, Bingzhe Wei, Kevin Chen-Chuan Chang, Yu Lei, and Bo Yang. Geom-gcn: Geometric graph convolutional networks. International Conference on Learning Representations (2019)

---

### Author Response · Authors · 2021-11-22
**General comment #2 (2021-11-22)**

We'd like to again thank the reviewers for engaging in helpful discussion. As discussed we have substantially expanded our experiment set, which we elaborate on below. We appreciate the feedback you've given on this and we agree that the fuller empirical support better supports the theory we propose, and so makes for a stronger paper. The changes in this revision can be found in Section 5 (page 8) and Appendix F (page 27).

In this revised version (2021-11-22) we have:
* Expanded our experiment set to include Chameleon, Squirrel, Actor, Citeseer and Pubmed alongside Cornell, Texas and Wisconsin;
* Added the +FA method from Alon et al 2020 as a benchmark;
* Extended our analysis on the effect on graph topology from just number of edges added/removed to also include plots of the degree distributions, as well as the Wasserstein distance (the transportation-based metric mentioned in the previous General Comment) between the original and transformed graphs;
* Included the measure of homophily from [1,2] in the results table, as well as restating the measure's definition in Appendix F;
* Included all hyperparameters chosen for all experiments in Appendix F.3.

The revised results table is in Section 5, at the top of page 8. We have also taken this opportunity to refactor our code for these experiments, which we will publish when possible. We have made sure to make the code modular so that our experiments are reproducible, as well as extendable if any future works would like to build off of it with new models or methods. We feel that the changes described in General Comments #1 and #2 based off of your feedback have made for a much stronger paper, and if the reviewer agrees, we hope that they would consider raising their score. Again we thank you for your responses and we continue to look forward to the coming discussion.

---

> ### Comment · Reviewer_kUJT · 2021-11-22
> **Score update**
>
> Thank you for the additional details and results.
>
> I have increased my rating to 10.
>
> Just note that Table 2 is now too wide and goes beyond the page's borders.

---

> ### Comment · Reviewer_Za5a · 2021-11-22
> **Reply**
>
> Thanks for the update! Performances of some datasets are not so significant. However, the overall paper makes a good contribution. I have increased my score.

---

### Decision · Program_Chairs · 2022-01-20

**Decision:**

Accept (Oral)

**Comment:**

The paper proposes a new technique to handle oversquashing in GNNs by introducing a novel rewiring technique. The reviewers are quite positive about the paper and the rebuttal phase greatly helped clarify the method and it's impact.